# Multicentennial record of Labrador Sea primary productivity and sea-ice variability archived in coralline algal barium

P. Chan[1], J. Halfar[1], W. Adey[2], S. Hetzinger[3], T. Zack[4], G.W.K. Moore[5], U.G. Wortmann[6], B. Williams[7] & A. Hou[1]

Accelerated warming and melting of Arctic sea-ice has been associated with significant increases in phytoplankton productivity in recent years. Here, utilizing a multiproxy approach, we reconstruct an annually resolved record of Labrador Sea productivity related to sea-ice variability in Labrador, Canada that extends well into the Little Ice Age (LIA; 1646 AD). Barium-to-calcium ratios (Ba/Ca) and carbon isotopes ($\delta^{13}$C) measured in long-lived coralline algae demonstrate significant correlations to both observational and proxy records of sea-ice variability, and show persistent patterns of co-variability broadly consistent with the timing and phasing of the Atlantic Multidecadal Oscillation (AMO). Results indicate reduced productivity in the Subarctic Northwest Atlantic associated with AMO cool phases during the LIA, followed by a step-wise increase from 1910 to present levels—unprecedented in the last 363 years. Increasing phytoplankton productivity is expected to fundamentally alter marine ecosystems as warming and freshening is projected to intensify over the coming century.

[1] Department of Chemical and Physical Sciences, University of Toronto, 3359 Mississauga Road North, Ontario, Canada L5L 1C6. [2] Department of Botany, Smithsonian National Museum of Natural History, 1000 Constitution Avenue, NW, Washington, District Of Columbia 20560, USA. [3] GEOMAR, Helmholtz-Zentrum für Ozeanforschung Kiel, Wischhofstr. 1-3, 24148 Kiel, Germany. [4] Department of Earth Sciences, University of Gothenburg, Guldhedsgatan 5A, 40530 Göteborg, Sweden. [5] Department of Physics, University of Toronto, 60 St George Street, Ontario, Canada M5S 1A7. [6] Department of Earth Sciences, University of Toronto, 22 Russell Street, Ontario, Canada M5S 3B1. [7] W.M. Keck Science Department, Claremont McKenna College, Pitzer College, Scripps College, 925 North Mills Avenue, Claremont, California 91711, USA. Correspondence and requests for materials should be addressed to P.C. (email: phoebetw.chan@utoronto.ca) or to J.H. (email: jochen.halfar@utoronto.ca).

The Subarctic North Atlantic is one of the most seasonally productive marine environments in the world, accounting for roughly 50% of global ocean productivity[1]. The seasonal melting of sea-ice in spring results in an increase in incident solar radiation, shoaling of the mixed layer, and release of nutrients and trace elements into the water column, triggering the onset of the spring phytoplankton bloom[2]. Phytoplankton abundance not only influences fisheries production and marine species diversity, but also plays an important role in atmosphere-ocean carbon exchange, and the export of carbon into the deep sea[3]. This blooming process is dependent upon physical properties of the surface seawater (for example, temperature, stratification, mixed-layer depth, light levels and nutrient availability) that are directly modified by climatological factors (for example, solar radiation, cloud cover, wind mixing and upwelling)[4].

Arctic sea-ice thickness and concentration have dropped by ~9% per decade since 1978 (ref. 5). Concurrent with this sea-ice decline is an increase in rates of phytoplankton productivity, driven by the enhanced transmittance of solar radiation into the surface ocean[6–9]. This ongoing loss in Arctic sea-ice cover has also been associated with an increased export of drift ice and freshwater out of the Arctic Ocean through the Fram Strait, and into the North Atlantic via the East Greenland Current[10] (Fig. 1a,b). Upon reaching the Davis Strait, the bulk of the transport is diverted westward and combines with the southward flowing polar waters originating from the Nares Strait and Canadian Arctic Archipelago to form the Labrador Current[11]. Since the mid-1960s, large pulses of polar freshwater (as evidenced by negative anomalies in ocean salinities) have been transported into the Nordic and Labrador Seas[12]. Therefore, continued warming of the surface ocean layer is expected to have significant impacts on primary productivity[13].

Phytoplankton growth in mid-to-high latitude regions is often limited by low light levels due to deep vertical mixing[13,14]. However, continued freshening associated with warming and sea-ice melt can supply additional buoyancy to the water column, reducing the mixed layer depth and leading to higher levels of productivity[8,14]. This has recently been confirmed by phytoplankton studies in Arctic and Subarctic basins that have revealed earlier timing[9,15], prolonged duration[6,9,16] and increased primary productivity[6,9,17] of the spring phytoplankton bloom. In fact, the most recent estimate indicates a 30% increase in net primary productivity in the Arctic Ocean between 1998 and 2012 (ref. 9). Enhanced biological productivity can significantly alter patterns of nutrient cycling in the ocean by essentially 'stripping' nutrients from the surface layer[18], and can also influence the life history and abundance of zooplankton (for example, *Calanus finmarchicus*) in the North Atlantic[19]; with implications for fishery yields and higher-level trophic interactions[16,20,21]. Thus, ocean warming, driving variations in sea-ice extent, volume and freshwater transport can play an important role in determining the timing, magnitude and duration of the spring bloom. However, difficulties of navigating in remote ice-laden waters and harsh polar climates have often resulted in short and incomplete records of *in situ* plankton abundance in the Northwestern Labrador Sea[22]. The Arrigo and van Dijken[9] satellite-derived productivity record (~15-year duration) represents the longest time series of primary productivity associated with changes in Arctic sea-ice cover, allowing for analyses of interannual oceanographic processes and their impacts on phytoplankton production to be made. While this represents a significant step forward in uncovering the driving mechanisms behind currently observed changes in ocean productivity, the relatively short record precludes analyses of long-term climatic variability before the period of satellite observation. Therefore, high-resolution reconstructions of primary productivity are needed, particularly in sparsely sampled regions such as the Northwest Atlantic, in order to place the currently observed productivity increase in the context of long-term climatic variability.

Carbon isotopes extracted from long-lived carbonate marine organisms have been commonly utilized as past recorders of surface water productivity in the North Atlantic[23–26]. However, this method is limited to productivity reconstructions before the second half of the last century due to an overprint by the anthropogenically induced decline of global carbon isotopes resulting from the burning of fossil fuels (Suess effect[27]). Alternatively, information of past ocean productivity may be gained through the study of trace nutrient distributions in the water column. Investigations of dissolved barium (Ba) concentrations in the Arctic reveal significant depletions of Ba in surface seawaters shortly following the spring phytoplankton bloom[28,29]. Barium is a naturally occurring biointermediate element that exhibits nutrient-type behaviour in the open ocean, characterized by depletions in the surface layer and enrichment with depth[28,30]. This pattern has been attributed to the uptake of Ba in the surface ocean associated with the formation of biological particulate matter, and the subsequent regeneration at depth due to the respiratory breakdown of organic matter[31–33]. Barium commonly behaves as a conservative dissolved tracer, and is incorporated into the carbonate skeleton of calcifying organisms in proportion to ambient seawater concentrations[34].

In this study, multicentennial records of barium-to-calcium trace element ratios (Ba/Ca) and carbon isotopic compositions ($\delta^{13}C$) were extracted from two long-lived annually banded specimens of the crustose coralline alga *Clathromorphum compactum,* as proxies for Labrador Sea primary productivity associated with climate-driven sea-ice variability. On the basis of this Ba/Ca proxy record, we examine how long-term climate oscillations driving sea-ice variability in the Subarctic Northwest Atlantic have influenced productivity since the Little Ice Age. Our annually resolved proxy record for surface ocean productivity shows a step-wise increase to levels unmatched over the last 363 years that is likely associated with warming ocean temperatures and sea-ice melt.

## Results

**Sample collection.** Two living specimens of the alga *Clathromorphum compactum* were collected: Sample Ki1 was collected in July 2011 at 15–17 m water depth, and sample 2013-15-4 in July 2013 at 17 m water depth via SCUBA off the east coast of Eastern Kingitok Island in Labrador, Canada ~15 km offshore from central Labrador, Canada (55°26′6.58″N, 59°51′55.57″W; Fig. 1a,b; for detailed description of sampling sites, see Adey *et al.*[35]. Additional information about crustose coralline algae can be found in Supplementary Note 1). Sample Ki1 was analysed for both magnesium-to-calcium (Mg/Ca) and barium-to-calcium (Ba/Ca) trace element ratios (363-year record), and specimen 2013-15-4 for annual Mg/Ca cycles and carbon isotopic composition ($\delta^{13}C$; 194-year record).

**Validation of the algal Ba/Ca productivity proxy.** Algal Ba/Ca ratios were correlated to an algal $\delta^{13}C$ record (commonly utilized as a measure of surface water productivity[27]), to validate the reliability of coralline algal Ba/Ca as a proxy for primary productivity. Oceanic $\delta^{13}C$ values declined steeply throughout the North Atlantic after 1960 due to anthropogenic inputs of isotopically light carbon[25]—which is unrelated to primary productivity. Post 1960 $\delta^{13}C$ data were thus excluded from the comparison and further analysis. Both the annually resolved and

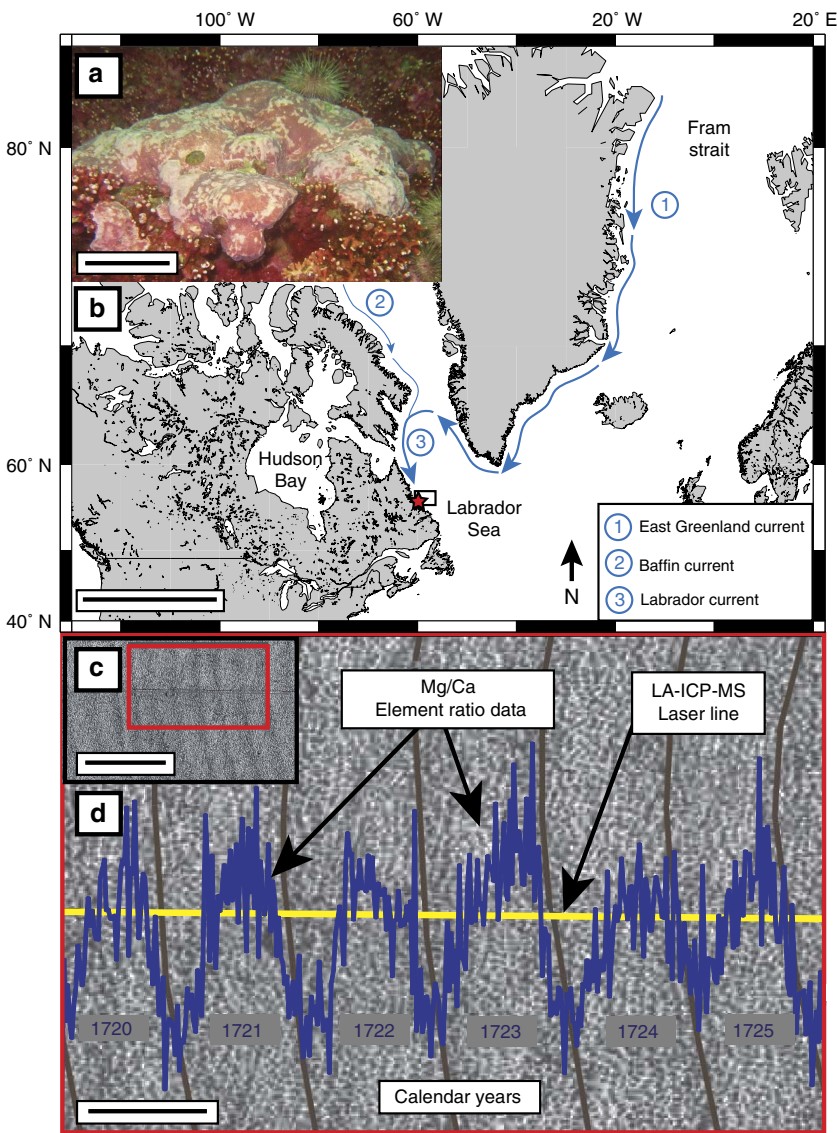

**Figure 1 | Crustose coralline alga in the Subarctic North Atlantic. (a)** Underwater image of *Clathromorphum compactum*. Scale bar, 20 cm. **(b)** Sample collection site off Kingitok Island in Labrador, Canada (red star). Retangular box shows region selected for spatial averaging of SeaWiFS ocean colour data and EN 4.2.0 gridded salinity data (56.0577–58.0133° N; 57.0905–61.0455° W). Arctic sea-ice and freshwater export follow major pathways of flow into the North Atlantic (blue arrows)—(1) East Greenland Current; (2) Baffin Island Current; and combine to form (3) Labrador Current. Scale bar, 2,000 km. Map created using open source software Generic Mapping Tools (GMT; available at: http://gmt.soest.hawaii.edu). **(c)** Image of cross-sectional polished specimen surface showing annual growth banding. Scale bar, 1 mm. **(d)** Enlarged image depicts assigned calendar years, growth increment boundaries (grey lines), LA-ICP-MS laser-line transect (yellow line) and Mg/Ca trace element ratio data (blue) plotted such that each Mg/Ca cycle represents 1 year of growth. Scale bar, 350 µm.

decadally smoothed algal Ba/Ca time series show a close correspondence to the algal $\delta^{13}C$ record from 1870 to 1960 (annual mean: $r = -0.53$, $P < 0.001$, $P_{adj} = 0.0038$; 10-year mean: $r = -0.89$, $P_{adj} = 0.003$; Fig. 2a–c). Significance levels were adjusted for loss in degrees of freedom using a lag-1 test that takes into account the temporal autocorrelation of the underlying time series[36] ($P_{adj} =$ adjusted for loss of degrees of freedom). Minor departures observed between the annually resolved algal Ba/Ca and $\delta^{13}C$ time series may be explained by natural variability within the proxy system, resulting from the records being obtained from two different specimens collected from nearby sites. The high degree of correlation between both the Ba/Ca and $\delta^{13}C$ time series before 1960 indicates that algal Ba/Ca ratios at this site can be used as a proxy for productivity that does not suffer from the anthropogenically induced carbon isotope

decline. Though it is possible for terrestrial runoff to overprint the barium productivity signal, oceanographic measurements of surface waters taken directly off the coast of Labrador show low barium and high salinity levels resulting from only small quantities of local freshwater input with low barium concentrations[37]. Correlations to gridded salinity data (Met Office Hadley Centre observations data set EN 4.2.0 (ref. 38)) for the region depicted in Fig. 1b ($56.0577 - 58.0133°$ N; $57.0905 - 61.0455°$ W), show no significant relationship to algal Ba/Ca between 1900 and 2009 on annual and decadal timescales (annual mean: $r = -0.19$, $P > 0.05$, $P_{adj} = 0.29$; 10-year mean: $r = -0.35$, $P_{adj} = 0.32$; Fig. 3a–c). This finding is in agreement with measurements of barium in the Baffin–Labrador region surface water, which indicate no significant relationships with sea surface salinity[37]. The sampling site is constantly bathed by the

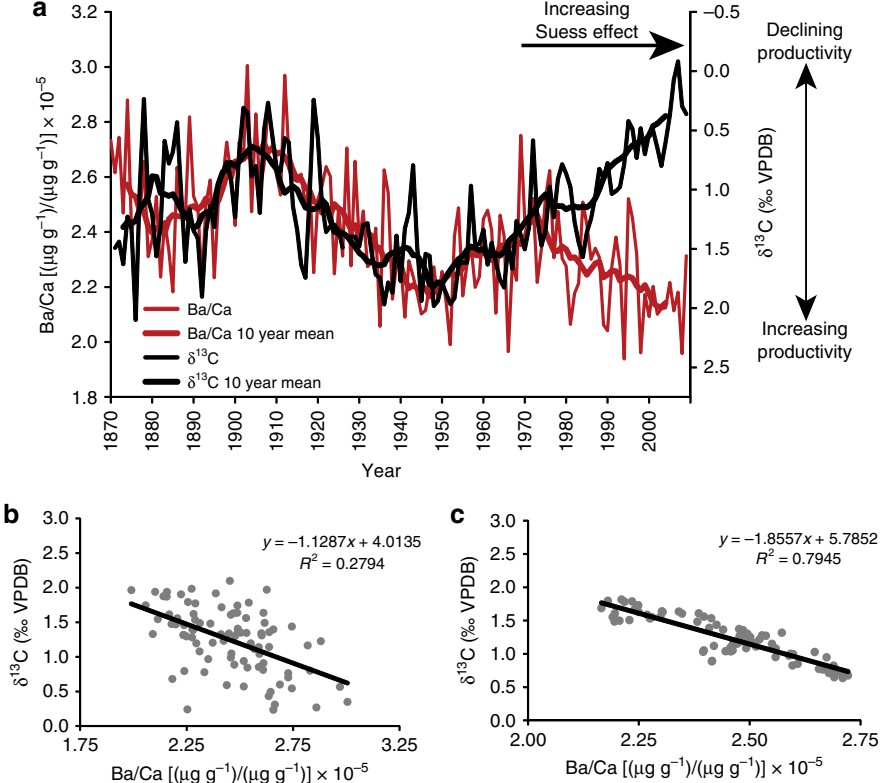

**Figure 2 | Relationship between algal Ba/Ca and $\delta^{13}$C.** (**a**) Annual and 10-year mean algal Ba/Ca ratios were derived from sample Kil and $\delta^{13}$C values from sample 2013-15-4 (plotted inversely on secondary x axis). The post-1960s departure to negative $\delta^{13}$C values is due to the anthropogenic introduction of isotopically light fossil fuels into the biosphere ('Suess Effect'). A $\delta^{13}$C decline starting ~1970 has commonly been observed in marine proxy archives throughout the North Atlantic[1]. Note that outliers defined as ± four times of interquartile range of raw $\delta^{13}$C values were removed from the comparison. Correlations between algal Ba/Ca and $\delta^{13}$C before the Suess Effect are displayed as scatter plots on (**b**) annual timescales (1870–1960; $r = -0.53$, $P < 0.001$; $P_{adj} = 0.0038$; s.e. of Ba/Ca ± 0.2e$^{-5}$); and (**c**) Decadal timescales (1870–1960; $r = -0.89$, $P_{adj} = 0.003$). Significance levels were adjusted for loss in degrees of freedom using a lag-1 test that takes into account the temporal autocorrelation of the underlying time series ($P_{adj}$ = adjusted for loss of degrees of freedom).

inshore branch of the southward-flowing Labrador Current, which, with a flow of 0.8 Sverdrup[11], dilutes any runoff from the numerous but small river systems along the Labrador coastline. In fact, the Labrador coastline experiences an average runoff of only 600–700 mm per year[39]. Therefore, the dominant source of barium depletion is likely to be associated with the drawdown of Ba through intense biological scavenging during primary production, rather than from minor terrestrial freshwater inputs off coastal Labrador.

**Algal Ba/Ca—sea-ice relationship.** The relationship between the algal Ba/Ca productivity proxy and regional sea-ice cover was tested by comparison to monthly Canadian Ice Service (CIS) historical sea-ice cover data to delimit the months of significant Ba/Ca—sea-ice cover relationships 1971–2009 (Fig. 4a). Autocorrelated results indicate that the months August to October of the ice-melt season positively correlate to algal Ba/Ca at the 90% level (grey bars), with the strongest correlations between August and September significant at the 95% and 99% levels, respectively (dark grey bars; Fig. 4a). Thus, algal Ba/Ca and monthly sea-ice data are most strongly coupled during months where ice-melt is most significant (August to October minimum sea-ice cover; solid black line). This also coincides with months exhibiting the warmest seawater temperatures recorded at depth off Kingitok Island (August to September)[35]. Coralline algae undergo seasonal growth cessation from late winter well into spring (January to May) in the Labrador coastal region. This is due to the presence of sea-ice and thick snow cover

which blocks sunlight for photosynthesis—leading to the depletion of overwintering stores of photosynthates with time under the sea-ice[35,40,41]. Therefore, the algae do not record environmental signals during these months.

Extending the algal Ba/Ca—sea-ice comparison to the full length of instrumental observations (since 1971), historical sea-ice coverage for northern Canadian waters was averaged seasonally according to the above established months of significant algal Ba/Ca—sea-ice relationships (August to October) and correlated to annual coralline algal Ba/Ca ratios (Fig. 4b; 1971 to 2009; $r = 0.57$, $P < 0.001$). Taking into account the temporal autocorrelation of the underlying time series, correlation was found to be statistically significant at the 99% confidence level ($P_{adj} = 0.013$). Hence, algal Ba/Ca ratios are related to late summer Labrador sea-ice coverage since the beginning of the observational data set in 1971. This may result from increased levels of biological scavenging following the seasonal ice breakup (effectively removing Ba from the surface water column), and reduces the availability of barium for uptake by the coralline algal skeleton. The above relationship is further supported by negative correlations between algal Ba/Ca and $\delta^{13}$C (Fig. 5a,b), and to spatially averaged chlorophyll $\alpha$ concentrations determined from Sea-Viewing Wide Field-of-View Sensor (SeaWiFS) ocean colour data (Fig. 1b; region in rectangle; 1998 to 2009; $r = -0.78$, $P = 0.0026$).

The Labrador Sea in the Subarctic Northwest Atlantic receives outflows of Arctic sea-ice and freshwater through the Canadian Arctic Archipelago and the Fram Strait, with the latter serving as the major gateway for polar outflows[10] (Fig. 1b). The Fram Strait represents the single largest conduit for export of sea-ice out of

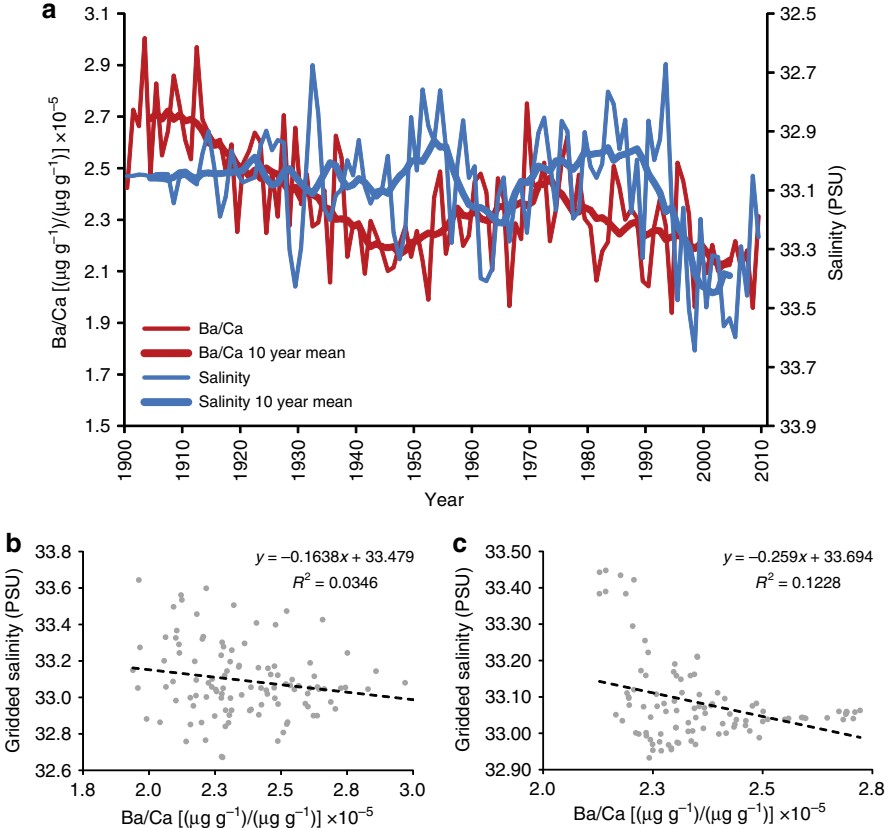

**Figure 3 | Algal Ba/Ca versus salinity.** (**a**) Gridded salinity data (Met Office Hadley Centre observations data set EN 4.2.0[2]) for the rectangular region depicted in Fig. 1b (56.0577–58.0133° N; 57.0905–61.0455° W) show no signficant correlations to algal Ba/Ca. Algal Ba/Ca versus salinity displayed as scatter plots on (**b**) annual timescales (1900–2009; $r = -0.19$, $P > 0.05$; $P_{adj} = 0.29$); and (**c**) Decadal timescales (1900–2009; $n = 101$, $r = -0.35$, $P_{adj} = 0.32$).

the Arctic, accounting for ~26% of the total sea-ice and freshwater discharge from the Arctic Ocean[42]. Global coupled atmosphere–sea-ice–ocean models demonstrate that during periods of anomalously large ice exports through the Fram Strait, greater volumes of ice are delivered southwards into the East Greenland Current, which subsequently melts and flows into the Labrador Sea as a freshwater/local sea-ice anomaly 1–2 years later[43,44]. These model simulations indicate that following large Fram Strait ice export events, anomalously cold and fresh waters enter into the Labrador Sea, with the ability to reduce salinity, deep convection, and ocean heat release—thus promoting local sea-ice formation[43,45]. To test whether sea-ice export through the Fram Strait influences local sea-ice formation (and hence productivity) in the Labrador Sea, coralline algal Ba/Ca was compared to an annually reconstructed record of Fram Strait sea-ice export from 1,870 to 2,000 (ref. 46). Correlations demonstrate a significant positive relationship between coralline algal Ba/Ca ratios and sea-ice export from the Fram Strait with a two-year lag (Fig. 5a,c; 1870–2000; 10-year mean: $r = 0.65$, $P_{adj} = 0.023$); the latter accounting for the transit time required for the Fram Strait sea-ice/freshwater signal to propagate into and form sea-ice in the Labrador Sea[43]. These results indicate that Fram Strait sea-ice export influences productivity in the Labrador Sea via local sea-ice formation, such that a decrease in Fram Strait sea-ice export leads to enhanced productivity, resulting in reduced levels of barium recorded in the coralline algal skeleton, and vice versa.

To examine whether the relationship between the algal multiproxy time series and ice-melt induced primary productivity is sustained over multicentennial timescales, both coralline algal Ba/Ca and $\delta^{13}$C were compared to a marine sea-ice proxy reconstruction[41] (Fig. 5a,d). This combined record of Mg/Ca trace element ratios with growth increment widths from crustose coralline algae has been used as a joint-proxy for the duration of the open water (ice-free) season associated with increased sea surface temperature (SST) and sunlight reaching the shallow seafloor[41]. The coralline algal Ba/Ca record correlates negatively to the combined proxy sea-ice record (Fig. 5a,d; 1646–2009; 10-year mean: $r = -0.75$, $P_{adj} < 0.001$), such that increases in sea-ice proxy values (related to warming, less sea-ice, and increased light levels) are associated with decreases in algal Ba/Ca (resulting from increased primary productivity). In contrast, pre-Suess effect carbon isotopes are positively related to the sea-ice proxy, and shows positive carbon isotope anomalies (increased productivity) during periods of reduced sea-ice (Fig. 5b,d).

**Productivity proxy linked to North Atlantic climate**. Historical records of North Atlantic sea-ice variability have recently been shown to covary with phase changes of the Atlantic Multidecadal Oscillation (AMO) index[47]. The AMO is manifested as basin-wide sea surface temperature anomalies in low to mid-latitudes (0–60° N), oscillating between positive and negative phases associated with warm and cool North Atlantic SSTs, respectively, with a periodicity of approximately 60–90 years. This multidecadal pattern of sea-surface temperature variability is thought to be driven by the varying intensity of the Atlantic Meridional Overturning Circulation (AMOC)[48]. Modelling simulations suggest that changes in the AMO and AMOC drive North Atlantic sea-ice extent, such that during periods of high

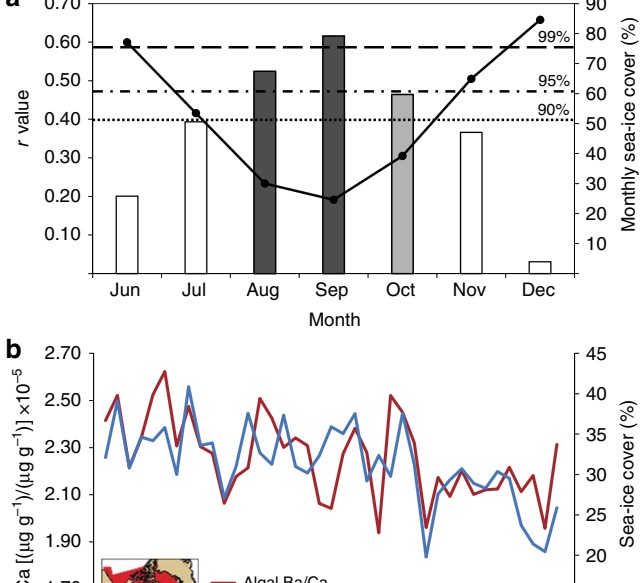

**Figure 4 | Relationship between algal Ba/Ca and Northern Canadian sea-ice cover.** (**a**) Correlation coefficient (r value) between annual Ba/Ca and monthly averaged sea-ice cover (bars), plotted against monthly averaged sea-ice cover (black line) from 1971 to 2009. Data are shown for months exhibiting significant algal growth (June to December)[40,41]. August to October shows significant correlations to algal Ba/Ca at the 95%, 99% and 90% levels, respectively (grey bars). White bars indicate insignificant correlations. (**b**) Annually averaged Ba/Ca correlated to seasonally averaged sea-ice cover for the months exhibiting significant sea-ice—algal Ba/Ca correlations (1971 to 2009; August to October; r = 0.57, P < 0.001, $P_{adj}$ = 0.013) for the region in red (inset).

index values (warm phases), intensification of deep convection leads to a subsequent release of heat that warms surface air and seawater temperatures in the Labrador and Nordic Seas, thus inhibiting the formation of sea-ice[49,50]. In contrast, periods of low AMO/AMOC index values (cool phases) have been linked to weakening of deep convection, enhanced sea-ice export through the Fram Strait, and an increased presence of sea-ice in the Subarctic North Atlantic[46,47,51].

Examination of the algal Ba/Ca time series reveals long-term multidecadal variability with a characteristic AMO-like frequency as illustrated using a multi-taper power spectrum on the gap-corrected time series (Fig. 6; 1646–2009; approximately 55–60 years significant at the 99% level). Multidecadal Ba/Ca cyclicities strongly covary with the instrumental AMO index[52] (Fig. 5a,e; 1900–2009; detrended 10-year mean: r = −0.83, $P_{adj}$ = 0.0028), such that AMO cool phases (1900–1927 and 1963–1995) are associated with higher Ba/Ca; and AMO warm phases (1928–1962 and mid-1990s to present) are associated with lower Ba/Ca levels.

On multicentennial timescales, coralline algal Ba/Ca continues to demonstrate an inverse relationship with a tree-ring proxy-based reconstruction of the AMO index[53] (Fig. 5a,f; 1646–1990; detrended 10-year mean: r = −0.55, $P_{adj}$ = 0.0016). This extended comparison between coralline algal Ba/Ca and the proxy-based AMO index reveals additional cycles of depressed algal Ba/Ca values that roughly correspond to the AMO warm phases (1670–1710 and 1780–1800), and increases in algal Ba/Ca

during AMO cool phases (1710–1735, 1745–1755, 1765–1780 and 1800–1820). Co-variability between the two proxy records suggests that more frequent periods of enhanced algal Ba/Ca peaks are associated with AMO cool phases during the Little Ice Age (1550–1850 AD).

## Discussion

Previously, Ba/Ca trace element ratios have been used for the reconstruction of coastal runoff, stratification, and/or upwelling depending on the source of barium in surface waters, which in turn is largely determined by the local setting of the sampling region[54–57]. As mentioned in the above, minor contributions of local runoff (characterized by low barium concentrations) are largely overprinted by the high-volume flow of the inshore branch of the Labrador Current. Thus, barium in this study is interpreted as a primary productivity signal. The Labrador shelf is characterized by moderate to high nutrient concentrations throughout most of the year[58]. Strong vertical mixing replenishes the supply of macronutrients to the surface, but reduces solar attenuation—the principal limiting factor on phytoplankton production in this region[58]. While it is possible for ice-melt induced freshening and stratification to limit nutrient upwelling, the consistently positive associations between algal Ba/Ca and sea-ice records indicate that the dilution of Ba does also not play a significant role in driving the variability observed in the algal Ba/Ca record. Instead, the positive sea-ice—algal Ba/Ca relationship reflects the biological scavenging of barium as a result of enhanced primary productivity related to the melting of sea-ice; in other words, a reduction of sea-ice cover is associated with higher levels of productivity, which in turn decreases barium availability for algal uptake. Increasing productivity has been linked to larger open water areas (providing suitable ice-free habitats for phytoplankton growth) and longer open water seasons (the timing between spring melt and fall freeze-up-which determines the length of the plankton growing season)[6]. Although no records of barium depletions associated with the recently observed increases in Arctic primary productivity currently exist; intensified biological productivity during seasonal phytoplankton blooms (related to increases in solar radiation and melting of sea-ice) have been shown to be capable of severely depleting and essentially 'stripping' barium from the surface seawater[28,29].

Before 1840, higher levels of algal Ba/Ca in the Labrador Sea coincide with the Little Ice Age (Fig. 5a), corresponding to lower levels of productivity associated with a cooler climate and extensive sea-ice. Between 1870 and 1910 the AMO shifted from a warm to cool phase, followed closely by a dip and subsequent rise in algal Ba/Ca that is mirrored by a decline in $\delta^{13}$C (Fig. 5a,b,f). This coincides with an increase in Fram Strait sea-ice export, and more extensive Labrador Sea ice cover as indicated by the sea-ice proxy time series (Fig. 5a,c,d). Multidecadal cyclicity observed in the various sea-ice and productivity records show a general synchrony with phase changes of the AMO; however, there are slight offsets in the peaks of the sea-ice proxy record (Fig. 5d). These apparent differences may be attributable to regional differences between the algal Ba/Ca record (obtained from a single location in Labrador, Canada) and the marine paleo sea-ice proxy reconstruction (averaged from different sites between Labrador and northern Baffin Island, Canada).

Based on satellite-derived estimates, ocean primary productivity has increased over the last 15 years. However, our multi-centennial algal Ba/Ca record displays a step-wise decline since 1910 which suggests that the recent instrumentally observed productivity increase actually began much earlier, in phase with declining sea-ice cover. While similar rates of decline have been

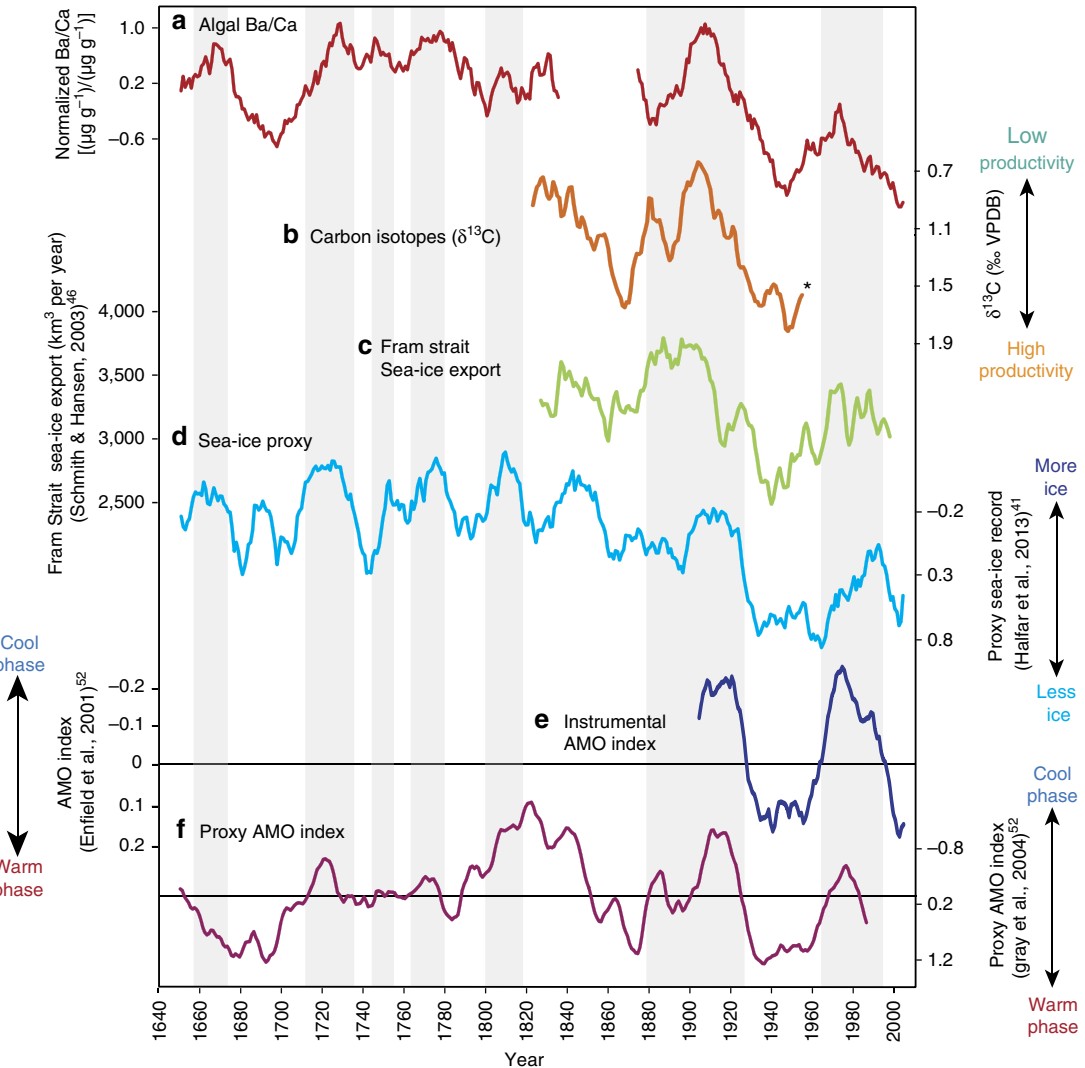

**Figure 5 | Coralline algal Ba/Ca compared to observational and proxy data.** Coloured lines represent 10-year means. (**a**) Coralline algal Ba/Ca was correlated to: (**b**) Coralline algal δ[13]C; (**c**) Fram Strait sea-ice export[46]; (**d**) Sea-ice proxy[41]; (**e**) Instrumental AMO index[52]; and (**f**) Tree-ring based proxy AMO index[53]. The algal δ[13]C time series in (**b**) was only plotted until 1960 (asterisk) due to overprinting by anthropogenic input of isotopically light carbon post 1960 (see main text for details). The full annually resolved δ[13]C time series is shown in Fig. 2a. Light grey bars indicate periods of higher Ba/Ca associated with increased sea-ice and cool phases of the Atlantic Multidecadal Oscillation. Note that (**d**) proxy sea-ice data, (**e**) instrumental AMO index, and (**f**) proxy AMO Index are plotted inversely to show increasing sea-ice and cooling.

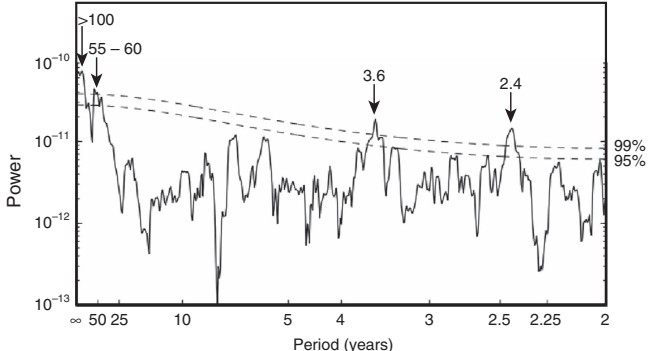

**Figure 6 | Multi-taper power spectrum of annually averaged gap-filled Ba/Ca time series 1646–2009.** Significance estimates based on red-noise AR (1) model shown in Supplementary Fig. 1. Periods (years) of peaks significant at 95/99% level are indicated.

observed in earlier parts of the algal Ba/Ca record (for example 1670–1700, 1910–1950), it is only recently that productivity has reached values exceeding those observed in the mid-twentieth century. These values signify the highest levels of productivity since the mid-seventeenth century, which may have resulted from the culmination of a strongly positive AMO state superimposed on the long-term twentieth century warming trend.

The algal Ba/Ca record presented here demonstrates that nutrient depletion (driven by intense phytoplankton blooms) is an important process controlling seawater Ba concentrations in highly productive regions such as the Labrador Sea. Synchronous relationships can be observed on multidecadal timescales between algal Ba/Ca (as a proxy for ocean productivity) and various instrumentally measured and proxy sea-ice records, all of which are related to phase changes of the AMO. Therefore, the here established climate-driven sea-ice—productivity—algal Ba/Ca relationship is likely to be a persistent feature of the Subarctic Northwest Atlantic.

The multicentennial record of coralline algal Ba/Ca ratios indicates that the recently observed productivity increase in the Subarctic Northwest Atlantic has reached levels unprecedented for the last 363 years. Only recently have these values exceeded those in the mid-twentieth century, which may be associated with a strongly positive AMO state further amplified by twentieth century warming. The ongoing loss of sea-ice leading to increasing levels of phytoplankton productivity is predicted to influence nutrient cycling patterns and carbon export to the deep sea, with the potential to fundamentally alter local marine ecosystems and affect fishery yields. This is particularly true for regions such as the Labrador Sea where there are relatively few trophic links in the food web, thus increases in primary productivity are likely to result in wholesale shifts in local species abundance and diversity.

## Methods

**Coralline alga preparation and scanning.** Algal samples were air-dried and vertically thick-sectioned into 3 mm-thick slabs along the axis of algal growth (perpendicular to growth increments), and sample surfaces were polished to a 3 μm finish. High-resolution digital photomosaics (a series of digital images stitched together to form a complete image) of the polished specimen surface were generated at the Department of Chemical and Physical Sciences at the University of Toronto using an Olympus reflected light microscope (VS-BX) attached to an automated sampling stage imaging system equipped with Geo.TS (Olympus Soft Imaging Systems) software (for details, see Hetzinger et al.[59]). This microscopic scanning setup enabled detailed viewing with zoom and panning functions using a pyramidal file format. The resulting photomosaics facilitated the establishment of precise age models through the clear identification of annual growth increments, which are delineated by couplets of dark and light bands (Fig. 1c; inset). Calendar years were assigned to each annual growth increment, starting from the year of collection and extending back in time at one year intervals (Fig. 1d). Digitized paths were traced along each annual growth increment boundary using Geo.TS for clear identification of growth layers. Transects for subsequent micromilling and Laser Ablation Inductively Coupled Plasma Mass Spectrometry (LA-ICP-MS) analysis were digitized along the growth axis of both samples also using Geo.TS. The laser-line transects were plotted in segments to minimize analytical drift, and allowed for the selection of the most pristine regions of the sample surface for laser analyses (for example avoiding areas of discontinuous growth due to bioerosion or breakage). In addition, laser-lines were overlapped by 3–5 years to ensure reproducibility and continuity of the barium record.

**LA-ICP-MS set-up.** LA-ICP-MS analysis of magnesium-to-calcium (Mg/Ca) and barium-to-calcium (Ba/Ca) trace element ratios were conducted at the Department of Earth Sciences at the University of Gothenburg using an Agilent 7,500a Quadrupole ICP MS attached to a ESI NWR213 laser ablation system equipped with a large format cell. Laser measurements were performed using laser energy densities of ca 7 J cm$^{-2}$, and 0.9 l He min$^{-1}$ as a carrier gas. 2 ml N$_2$ min$^{-1}$ were added downstream to the He carrier gas before mixed with 0.7 l min$^{-1}$ Ar. The addition of N$_2$ along with the use of a second rotary vacuum pump lead to an improvement in sensitivity by a factor of two compared to previous measurements (for example Chan et al.[57]). Transect lengths were limited to 10,000 μm in order to minimize the effect of ICP-MS drift and were analysed at a scan speed of 10 μm s$^{-1}$, with a 60 μm spot size and laser frequency of 10 Hz. NIST SRM 610 (US National Institute of Standard and Technology Standard Reference Material) glass reference material was used as an external standard. NIST SRM 610 concentrations used are 81,595 p.p.m. for Ca, 435 p.p.m. for Mg and 452 p.p.m. for Ba, extracted from the GeoReM database (available from: http://georem.mpch-mainz.gwdg.de/, version 04/01/2012). Detection limits were: Mg = 0.01 p.p.m., Ca = 4 p.p.m., Ba = 0.005 p.p.m. All data are reported as Element/Ca mass ratios (($\mu$g g$^{-1}$)/($\mu$g g$^{-1}$)), which can be converted to molar ratios (mol mol$^{-1}$) by dividing ratios by a conversion divisor (0.60644 for Mg/Ca, 3.42649 for Ba/Ca)[37].

**Age model development.** Subannually resolved age models were established based on seasonal cyclic variations in Mg/Ca element ratios. While LA-ICP-MS derived Mg/Ca cycles were available for sample Ki1, qualitative Mg/Ca cycles were obtained in the W.M. Keck Science Department of Claremont McKenna College, Pitzer College, and Scripps College using a Perkin Elmer 8,300 ICP OES attached to an ESI NWR213 LA system equipped with a large format cell for the purpose of establishing an age model for sample 2013-15-4. According to monthly averages of Extended Reconstructed Sea Surface Temperatures (ERSST v4; 1854–2009 (ref. 60)), the maximum (minimum) Mg/Ca values of seasonal cycles were interpreted to correspond to months exhibiting the warmest—August (coolest—March) temperatures off Kingitok Island, Labrador. Maximum and minimum Mg/Ca values were determined by graphically superimposing seasonal Mg/Ca cycles onto a photomosaic of the sample surface depicting laser transects, with the

maxima of Mg/Ca aligned to the central portion of each growth increment, and the minima aligned to growth increment transitions (Fig. 1d). This cross-checking method ensured that each individual Mg/Ca cycle was matched with a corresponding annual growth increment in order to avoid possible errors in the age model. Anchor points were assigned beginning with the first Mg/Ca minimum (for example March 2010), followed by the subsequent Mg/Ca maximum (for example August 2009) and continuing back in time. The algal Mg/Ca time series for both samples were then linearly interpolated between these anchor points using Ana-lySeries software[61] to obtain an equidistant proxy time series at a resolution of 12 samples per year. Mg/Ca cycle based age models were confirmed by radiocarbon dating, where all ages obtained by cycle counting fell within the error of the calibrated radiocarbon age ranges[42]. As sample collection took place in the summer, the final year of algal growth and the outermost layer before collection was incomplete and therefore excluded from the analyses. Last, since algal Ba/Ca ratios do not exhibit annual cyclicity, Mg/Ca anchor points were transferred to barium laser measurements in order to create the Ba/Ca age model.

A 30-year hiatus in the algal Ba/Ca time series is evident from 1840 to 1870 as a result of a gap in LA-ICP-MS measurements. LA-ICP-MS transects were taken parallel to previously measured electron microprobe lines extending across the entire lifespan of the algal specimen[41]. LA-ICP-MS analysis was conducted in order to obtain Ba/Ca records from the algal specimen as microprobe analyses were limited to measuring only Mg/Ca ratios. Through electron microprobe analyses, a geochemically altered region of the specimen was discovered (as indicated by significantly depressed amplitudes of Mg/Ca cycles), and was therefore avoided for subsequent laser analyses. However, detailed microscopic inspection of the sample surface within the altered region displayed clear growth banding, which enabled precise calendar dating of the period where no laser measurements were taken. These dates were further confirmed using radiocarbon ages, which indicated that there was no significant hiatus in the altered region[41]. The age model for algal Ba/Ca was then developed accordingly to account for the years of missing data in the altered region. A detrended Ba/Ca record was used for correlations to the proxy AMO index. Before detrending, a Singular Spectral Analysis (SSA[62]) was performed in order to fill the gap in laser data collection. This gap-filling technique ensured that the linear trend in the gap-filled time series was not significantly different than the unfilled time series (significant at the 99% level; Supplementary Fig. 1).

**Carbon isotope analysis.** Material for isotope analysis was obtained from specimen 2013-15-4 by micromilling of 247 samples parallel to growth increments using a milling path spacing of 180 μm (for details on micromilling technique applied to coralline algae see Williams et al.[63]). About 200 μg of material was introduced in 5 ml vacutainers, and all ambient atmosphere was flushed with a helium stream. Five to ten drops of phosphoric acid was added to the sample vial and heated at 70 °C for 1 h before analysis. The evolving CO$_2$ was analysed for $\delta^{13}$C using a Thermo Finnigan Gas Bench II coupled to a Thermo Finnigan Mat 253 isotope ratio monitoring mass spectrometre in continuous flow mode. The system was calibrated using the following standards CaCO$_3$ Merck, IAEA-CO-8, IAEA-CO-1 and NBS 19. Analytical reproducibility was determined from repeated measurements at 1-sigma = ± 0.07‰ for $\delta^{13}$C. Samples were obtained for the last 194 years of growth of specimen 2013-15-4 according to an age model established from annual Mg/Ca cycles. Based on this age model $\delta^{13}$C values from the 247 samples measured were downsampled to annual resolution (194 years) using AnalySeries software[61].

**Observational and proxy data sets.** Historical ice coverage data from northern Canada was obtained from the Canadian Ice Service, Environment Canada (https://www.ec.gc.ca/glaces-ice/). Percentage ice cover data in the northern Canadian waters region was obtained from IceGraph Tool 2.0 for the months exhibiting the minimum sea-ice cover (August to October) from 1971 to 2009 (ref. 64). The data from the IceGraph Tool 2.0 are derived from weekly regional ice charts from predefined regions (based on the compilation analysis of satellite imagery, weather and oceanographic information, and observations from ships and aircrafts), and is calculated by multiplying the percentage of sea area in a specified region that is covered by ice, by the concentration of ice. Chlorophyll α concentrations (mg m$^{-3}$) for the years 1998–2009 were determined from Level 3 (monthly, 9 km resolution) Sea-Viewing Wide Field-of-View Sensor (SeaWiFS) ocean colour data obtained using the Giovanni online data system, developed and maintained by the NASA Goddard Earth Sciences Data and Information Services Center (GES DISC; http://gdata1.sci.gsfc.nasa.gov/daac-bin/G3/gui.cgi?instance_id=ocean_month). Fram Strait sea-ice export data were obtained from a reconstruction based on historical observations of multiyear ice called 'Storis' from ship logbooks and ice charts off the southwestern coast of Greenland[46]. Since summer months contain the most data coverage, summer storis observations extracted from historical records were used to create the storis extent index[46] (May to July; 1820–2000). Although the Fram Strait sea-ice export record dates back to the early 1800s, decadal-scale relationships to algal Ba/Ca (based on 10-year running means) were only examined for the period from 1870 to 2009 due to the gap in LA-ICP-MS analysis of the algal Ba/Ca record. Similarly, correlation coefficients between $\delta^{13}$C and Ba/Ca ratios were only determined for the post-gap period until 1960, when

the Suess effect resulted in a significant anthropogenic overprinting of the $\delta^{13}C$ signal.

**Data availability.** All relevant data are available upon request from the authors. Requests for materials should be addressed to Phoebe Chan (phoebetw.chan@utoronto.ca).

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

## Acknowledgements

We thank Thew Suskiewicz for assisting with algal collections and Anne de Vernal for stimulating discussions. We acknowledge funding from the Natural Sciences and Engineering Research Council of Canada to J.H. and P.C. and the Geological Society of America to P.C.

## Author contributions

P.C. analysed and interpreted the data, and wrote the manuscript with J.H. who conceived and supervised this research project. W.A. collected algal specimens, and provided background on coralline algal ecology. S.H. aided in interpretation of the results and provided statistical guidance for this project. T.Z. conducted LA-ICP-MS analyses. G.W.K.M. performed the Singular Spectral Analysis (SSA) and multi-taper power spectrum on the time series. U.G.W. and A.H. analysed and interpreted carbon isotope data. B.W. conducted LA-ICP-OES analyses, and all coauthors provided feedback on the manuscript.

## Additional information

**Competing interests:** The authors declare no competing financial interests.

