## [Peer Review File · Nature Communications]

Reviewers' comments:

Reviewer #1 (Remarks to the Author):

A. Overview and Summary of the key results

This paper investigates historic marine productivity using reconstructions from coralline algal derived via sea ice extent. The paper makes some interesting links between past sea ice and productivity (planktonic and fish) which is a major research question and I commend the authors on this. However, while the Ba/Ca reconstruction is robust, linking it to productivity at centennial time scales is not as simple as indicated (see below). Also, the paper is quite hard to follow after the intro (which is well written) due to the structure and very short discussion.

B. Originality and interest: if not novel, please give references

The approach is certainly novel at this time scale. Decadal length productivity is available from satellite data.

C. Data & methodology: validity of approach, quality of data, quality of presentation

Productivity is not only dependent on the extent of sea ice but also the availability of nutrients (e.g. nutrient upwelling). Thus it is a combination of sea ice extent and nutrient availability that drive productivity. At short temporal scales (e.g. cruises and satellite-based productivity), productivity estimates can integrate sea ice extent and nutrient availability. However, ice extent and upwelling are not simply linked and are certainly not temporally stable. So reconstructing sea ice extent does not necessarily account for nutrient availability (e.g. excessive fresh water ingress may stop nutrient upwelling) and would thus not give a real measure of productivity (e.g. see Frey, K. E. et al., 2014). Significantly, the coralline algae came from the coast, that productivity will be a minor proportion of the sea ice - nutrient driven productivity that occurs in the open ocean.

The sample is collected coastally which is affected by fresh water runoff. The authors demonstrate Ba is also affected by fresh water (page 7 "In order to test whether freshwater and sea-ice export"). Without separating the freshwater signal driving Ba from the productivity signal driving Ba how can marine productivity be reconstructed?

Page 6 sentence starting "The positive sea-ice - algal Ba/Ca relationship reflects the biological scavenging of barium from the surface seawater.....". Ba is also associated with fresh water / salinity (as indicated in the introduction and the results) so having this three stage relationship between sea ice melt - productivity - algal Ba concentrations is not accurate. To directly link algal Ba/Ca concentration with productivity, the authors should calibrate the algal Ba/Ca with productivity and fresh water separately.

Page 7: "Coralline algal Ba/Ca was compared to a marine sea-ice proxy reconstruction⁴² and a century-long observation of winter sea-ice extent off the coast of Newfoundland⁴³ (Figs 3a-c), in order to examine whether the algal barium relationship to ice-melt induced primary productivity is sustained over longer timescales." and similar statements throughout. As above, this does not mean productivity is the only variable affecting Ba.

D. Appropriate use of statistics and treatment of uncertainties

Page 6: "demonstrates a strong negative relationship (1998 - 2009; n = 12, r = 0.74, p =

0.0055)." The r value is positive not negative. This is key to interpretation of the data, check if negative or positive?

Other statistics appear appropriate

E. Conclusions: robustness, validity, reliability

No conclusions are included in the paper.

F. Suggested improvements: experiments, data for possible revision

As above, a separate calibration for productivity is needed.

G. References: appropriate credit to previous work?

Yes

H. Clarity and context: lucidity of abstract/summary, appropriateness of abstract, introduction and conclusions

The text does not flow, particularly in the results, there results are interspersed with methods and discussion. Also the discussion is very short, relevant parts should be moved from the results.

Introduction is well written.

Results: first paragraph in the results refers to methods, this should be labelled as such.

Fig 2 legend includes some results, these are better placed in the results section.

Page 5 last line: This sentence should be in the methods. "a test", give the name of the test.

Page 6 is a single paragraph, this should be sectioned into smaller paragraphs

Page 6 "enhanced productivity related to the melting of sea-ice," I think this refers to coralline algal productivity? Please specify.

Page 6: sentence starting "This finding is in agreement with a number of Arctic-wide investigations....." This should be in the discussion

Page 11: Paragraph starting "A 500-year long historical record of North Atlantic cod landings off Newfoundland shows that stocks remained low throughout the Little Ice Age (LIA; 16th - late 19th century)56.)". This is too speculative, I suggest you remove the paragraph.

Discussion: first paragraph. Without separating the impact of freshwater, these assumptions are hard to square.

Reviewer #2 (Remarks to the Author):

REVIEW of Chan et al. "Multicentennial Record of North Atlantic Primary Productivity and Sea-Ice Variability Archived in Coralline Algal Ba/Ca"

A. Summary of the key results. The paper presents results on an important aspect of natural modes of variability that are superposed upon anthropogenic global warming. The existence of an apparently persistent broad relationship between marine primary productivity and the Atlantic Multidecadal Oscillation (AMO)/sea-ice-related mode is valuable knowledge. These results are from the development and analysis of a multi-century high-resolution proxy indicator of primary Productivity - this is new and notable.

B. Originality and interest: The paper is similar in scope and approach - and is based on material obtained from the same site as - to the Halfar et al. (2013) in Proc. Natl. Acad. Sci., but adds significantly to that paper. Here the focus is on developing a proxy for primary productivity and demonstrating its broad relationship to the sea-ice cover and the AMO.

Given that the originality, the significance of the results and broad biogeochemistry interest factor are all high, I would like to see this published in Nature Commun., pending however some revisions that are important enough to be considered "major" - even though addressing/correcting these shortcomings can be done with a minor amount of effort. These issues are concisely described below in Points (1-4).

(1) Saying that this represents primary productivity in the North Atlantic is an over-statement. This is based on a site at the very margins of broad and diverse region. And while the SeaWiFS extrapolation takes us from site-specificity to a small region (again at the margins of the northwest subarctic North Atlantic)- and there are correlations with the AMO and regional sea ice data records - the results from this coral specimen from a marginal site cannot be considered to represent the North Atlantic. Because the title is already as long and specific as it can be, then at least qualify this aspect prominently in the paper, certainly in the abstract as well as the conclusions.

C. Data & methodology: The data and methods are appropriate, and are reasonably well presented in the main manuscript and in the Suppl. Information.

One aspect that needs to be clarified is:

(2) There are differences apparent in the various sea ice data records (Newfoundland sea ice extent, Fram Strait sea ice export, and the paleo proxy of Halfar et al.) While all show multidecadal variability, there are differences, e.g., in mid-to-late 20th century, where the Ba/Ca peak doesn't appear to correspond well with the Newfoundland ice or the paleo proxy sea ice indicator (Fig. 3 a, b, c). It does however match well with the Fram Strait sea ice export (Fig. 3 d). The authors need to further address this issue, and offer an explanation - maybe the Newfoundland record and/or coral-based sea-ice proxy does not closely reflect sea ice in parts of the record. It is presumably not an issue of chronology nor resolution, as these records are all annual or better - or is there something going on in the Ba/Ca record in that respect?

D. Statistics and treatment of uncertainties: There is well-reasoned and correct application of statistics, including correlation analysis that is adjusted for considerable temporal autocorrelation in each of the data records compared. The SSA and the frequency domain spectral analysis appear correct including confidence Levels.

E. Conclusions: The robustness, validity and reliability depends to some degree on addressing points (1) and (2) above.

F. Suggested improvements: Specific issues in addition to the points above:

(3) Pg. 2: "This ongoing loss in Arctic sea-ice cover has also led to an increased export of drift ice

and freshwater out of the Arctic Ocean through the Fram Strait." This is not necessarily true as stated. There may be an association between the two, but the relationship can even be the opposite to this statement: that is, an increase in export of sea ice from the Arctic Ocean through the Fram Strait can itself lead to decreased sea-ice cover in the Arctic. Case in point: the then-record sea ice minimum in 2007 that was largely due to anomalously high ice export.

(4) Pg. 12. "This multicentennial record of coralline algal Ba/Ca ratios indicates that the recently observed productivity increase in the Subarctic North Atlantic is unprecedented in the last 365 years." The indicator of primary productivity does indeed have its highest value in the most recent years, but the increase (rate of change) is not unusual, so please re-phrase. And it could be mentioned that only very recently have the values exceeded those seen in the mid-20th century, associated with the culmination of the Early 20th Century Warming and a positive AMO state.

G. References: The Reference list is appropriate and excellent. No additions or deletions recommended.

H. Clarity and context: The presentation quality is consistently high, including the abstract, introduction and conclusions. The figures are also appropriate in number, information content and are of high quality.

Reviewer #3 (Remarks to the Author):

The manuscript "Multicentennial Record of North Atlantic Primary Productivity and Sea-Ice Variability Archived in Coralline Algal Ba/Ca" by Chan et al. is a very well written description of a paleo-proxy study of changes in primary production in the Arctic over the last 500 years. I am not a paleo-oceanographer so I won't comment on the methods used, although they appear to be consistent with other papers I have read on similar topics. The results are very interesting and important and well worth publishing in Nature Communications.

I only have a few comments that may need to be addressed by the authors.

It is not clear how solidly it has been established that the Ba/Ca ratio is a proxy for ocean production. I understand the logic employed, but it would be useful for the authors to provide some recent evidence that this proxy has been used reliably in the way the authors use it. The most recent of the three papers they cite is from 1988. I would expect that a good proxy would have been used more often and more recently than that.

Chlorophyll is not the same as primary production and should not be treated as such. This is especially important for their study since increases in primary production in recent years have been attributed to lower sea ice cover and longer growing seasons, not to higher chlorophyll concentrations. Phytoplankton are growing for a longer period of time, but not necessarily attaining higher biomass.

I was also left wondering if the growth of coralline algae itself can affect the Ba/Ca and Mg/Ca ratios. This was not addressed directly, although the authors allude to the fact that they record seawater ratios accurately. Also, the authors note a negative correlation between Ba/Ca (a proxy for production) and Mg/Ca (a proxy for ice cover) ratios. Considering these two proxies are clearly not independent (both come from the coralline algae and both contain Ca), is this a valid exercise? It would also be useful for the authors to provide the basis for using Mg/Ca as a proxy for ice cover.

Detailed point-by-point response to referees:

Reviewer #1 (Remarks to the Author):

A. Overview and Summary of the key results

This paper investigates historic marine productivity using reconstructions from coralline algal derived via sea ice extent. The paper makes some interesting links between past sea ice and productivity (planktonic and fish) which is a major research question and I commend the authors on this. However, while the Ba/Ca reconstruction is robust, linking it to productivity at centennial time scales is not as simple as indicated (see below). Also, the paper is quite hard to follow after the intro (which is well written) due to the structure and very short discussion.

Using an additional long-lived algal sample from the same collection site we generated an annually-resolved, 194-year carbon isotope record. Carbon isotopes in calcium carbonate marine organisms have been used in numerous studies as proxies of marine productivity, and are therefore well established. We demonstrate that both the algal Ba/Ca and carbon isotope records are closely related during the period of overlap (1819 – 1960), which is strong evidence that Ba/Ca ratios in the present study can be utilized as a productivity proxy. Carbon isotope ratios globally are now overprinted by the anthropogenic emissions of isotopically light carbon from fossil fuels. This well-known “Suess effect” is becoming evident in many Atlantic carbon isotope time series since the 1960s, making the carbon isotope productivity proxy problematic for recording changes in productivity over the last 50 years. Hence, Ba/Ca ratios (which do not suffer from a Suess-effect) are a promising alternative for studying productivity in certain settings.

The above information has been included in different sections of the manuscript in lines: 119 – 133, 191 – 203, 264 – 268, and 394 – 408. Comparisons between Ba/Ca and carbon isotopes as productivity proxies are displayed in Fig. 2 a, b as 10-year means (1819-1960), and in Supplementary Fig. 1 as annually-resolved time series for the entire duration of both records, illustrating the influence of the Suess effect on carbon isotopes over the last five decades.

The structure of the manuscript has been modified to better separate the results from the discussions, and at the same time expanding the discussion, and adding a conclusions section.

B. Originality and interest: if not novel, please give references

The approach is certainly novel at this time scale. Decadal length productivity is available from satellite data.

C. Data & methodology: validity of approach, quality of data, quality of presentation

Productivity is not only dependent on the extent of sea ice but also the availability of nutrients (e.g. nutrient upwelling). Thus it is a combination of sea ice extent and nutrient availability that drive productivity. At short temporal scales (e.g. cruises and satellite-based productivity), productivity estimates can integrate sea ice extent and nutrient availability. However, ice extent and upwelling are not simply linked and are certainly not temporally stable. So reconstructing

sea ice extent does not necessarily account for nutrient availability (e.g. excessive fresh water ingress may stop nutrient upwelling) and would thus not give a real measure of productivity (e.g. see Frey, K. E. et al., 2014).

The Labrador shelf is characterized by moderate to high nutrient concentrations for most of the year. Strong vertical mixing replenishes the supply of macro-nutrients to the surface, but reduces solar attenuation - the principal limiting factor on phytoplankton production in this region.

While excessive freshwater ingress or sea-ice induced stratification may limit nutrient upwelling, this would result in a negative relationship between sea-ice and Ba. However, in this study we observe a statistically significant positive relationship between algal Ba/Ca with both observational and proxy sea-ice proxy data, which demonstrates that the freshwater related to sea-ice melt is not likely to be diluting the nutrients.

This is now included in the discussion in lines: 245 – 251.

Significantly, the coralline algae came from the coast, that productivity will be a minor proportion of the sea ice - nutrient driven productivity that occurs in the open ocean.

The coralline algae were collected on a near coastal island located on the Labrador shelf approximately 23 km away from the Labrador Coastline. This has now been clarified in the revised manuscript in lines: 110 – 114.

The sample is collected coastally which is affected by fresh water runoff. The authors demonstrate Ba is also affected by fresh water (page 7 PAGE 9 "In order to test whether freshwater and sea-ice export"). Without separating the freshwater signal driving Ba from the productivity signal driving Ba how can marine productivity be reconstructed? Declining Ba could be increasing FW and not PP??

There are no major river systems along the Labrador coastline near the (offshore) collection site, thus significant terrestrial freshwater runoff (which may dilute nutrients) can be considered negligible. Therefore, the scavenging and removal of nutrients via primary productivity is likely the dominant signal in this region. This has been included in the revised manuscript, in lines: 129 – 133 and 248 – 251.

In addition, as outlined above, the comparison of algal Ba/Ca to the carbon isotope record clearly demonstrates the relationship with primary productivity.

Page 6 sentence starting "The positive sea-ice - algal Ba/Ca relationship reflects the biological scavenging of barium from the surface seawater.....". Ba is also associated with fresh water / salinity (as indicated in the introduction and the results) so having this three stage relationship between sea ice melt - productivity - algal Ba concentrations is not accurate. To directly link

algal Ba/Ca concentration with productivity, the authors should calibrate the algal Ba/Ca with productivity and fresh water separately.

As described above, we argue that the site is a) not coastal, and b) there is little freshwater input along the portion of the Labrador coastline that is close to the algal collection site. Furthermore, in this revised version we have included carbon isotopes as additional evidence for using Ba/Ca as a productivity proxy as outlined above.

Page 7: "Coralline algal Ba/Ca was compared to a marine sea-ice proxy reconstruction and a century-long observation of winter sea-ice extent off the coast of Newfoundland (Figs 3a-c), in order to examine whether the algal barium relationship to ice-melt induced primary productivity is sustained over longer timescales." and similar statements throughout. As above, this does not mean productivity is the only variable affecting Ba.

There are certainly a number of other factors influencing the source and supply of Ba into the surface ocean (not the main focus of this paper), but primary productivity is likely the dominant cause for Ba depletion in this biologically productive region. This is shown by the additional evidence added to this revised manuscript (see above).

D. Appropriate use of statistics and treatment of uncertainties

Page 6: "demonstrates a strong negative relationship (1998 - 2009; $n = 12$, $r = 0.74$, $p = 0.0055$)." The r value is positive not negative. This is key to interpretation of the data, check if negative or positive? Other statistics appear appropriate

Thank you for noticing this mistake! A negative sign has been added to the r value (line 166).

E. Conclusions: robustness, validity, reliability

No conclusions are included in the paper.

A conclusions section was added to the manuscript (lines 287 – 305).

F. Suggested improvements: experiments, data for possible revision

As above, a separate calibration for productivity is needed.

A separate calibration for Ba as a proxy for productivity was made using carbon isotope record. See detailed statements above.

G. References: appropriate credit to previous work?

Yes

H. Clarity and context: lucidity of abstract/summary, appropriateness of abstract, introduction and conclusions

The text does not flow, particularly in the results, the results are interspersed with methods and discussion. Also the discussion is very short, relevant parts should be moved from the results. Introduction is well written.

The overall structure of the manuscript has been modified. Material pertaining to methods (ex. sample preparation and scanning) was moved into the methods section at the end of manuscript. The discussion was rewritten and expanded with relevant parts moved from the results.

Results: first paragraph in the results refers to methods, this should be labelled as such.

The sample preparation and scanning information from the first paragraph of the results were moved to the methods section (lines 308 – 327).

Fig 2 legend includes some results, these are better placed in the results section.

The results in Fig. 2 legend (now Fig. 3 in revised manuscript) have been moved into the results section (lines 135 – 151).

Page 5 last line: This sentence should be in the methods. "a test", give name of test.

The name of the test is simply called the lag-1 test that adjusts for loss in degrees of freedom by taking into account the temporal autocorrelation of the underlying time series. This was included in lines 139 – 140 of the revised manuscript.

Page 6 is a single paragraph, this should be sectioned into smaller paragraphs

Page 6 paragraph has been sectioned into smaller paragraphs, with relevant sections moved into the discussions (lines 163 – 166, 251 – 262).

Page 6 "enhanced productivity related to the melting of sea-ice," I think this refers to coralline algal productivity? Please specify.

This statement refers to primary productivity. Changes have been made to clarify the sentence: “Instead, the positive sea-ice – algal Ba/Ca relationship reflects the biological scavenging of barium from the surface seawater as a result of enhanced primary productivity related to the melting of sea-ice ...” (lines 251 – 255).

Page 6 sentence starting "This finding is in agreement with a number of Arctic-wide investigations....." This should be in the discussion

The sentence has been reworked and moved to the discussion (line 277 – 278).

Page 11 Paragraph starting "A 500-year long historical record of North Atlantic cod landings off Newfoundland shows that stocks remained low throughout the Little Ice Age (LIA; 16th - late 19th century)56.)". This is too speculative, I suggest you remove the paragraph.

This paragraph has been removed in the revised manuscript.

Discussion: first paragraph. Without separating the impact of freshwater, these assumptions are hard to square.

As stated above, the arguments using the carbon isotope record and the fact that this site is distant to the coastline and receives little freshwater input, provide strong evidence that the freshwater impact is negligible.

Reviewer #2 (Remarks to the Author):

REVIEW of Chan et al. "Multicentennial Record of North Atlantic Primary Productivity and Sea-Ice Variability Archived in Coralline Algal Ba/Ca"

***A. Summary of the key results.** The paper presents results on an important aspect of natural modes of variability that are superposed upon anthropogenic global warming. The existence of an apparently persistent broad relationship between marine primary productivity and the Atlantic Multidecadal Oscillation (AMO)/sea-ice-related mode is valuable knowledge. These results are from the development and analysis of a multi-century high-resolution proxy indicator of primary Productivity - this is new and notable.*

***B. Originality and interest:** The paper is similar in scope and approach - and is based on material obtained from the same site as - to the Halfar et al. (2013) in Proc. Natl. Acad. Sci., but adds significantly to that paper. Here the focus is on developing a proxy for primary productivity and demonstrating its broad relationship to the sea-ice cover and the AMO.*

Given that the originality, the significance of the results and broad biogeochemistry interest factor are all high, I would like to see this published in Nature Commun., pending however some revisions that are important enough to be considered "major" - even though addressing/correcting these shortcomings can be done with a minor amount of effort. These issues are concisely described below in Points (1-4).

(1) Saying that this represents primary productivity in the North Atlantic is an over-statement. This is based on a site at the very margins of broad and diverse region. And while the SeaWIFs extrapolation takes us from site-specificity to a small region (again at the margins of the northwest subarctic North Atlantic)- and there are correlations with the AMO and regional sea ice data records - the results from this coral specimen from a marginal site cannot be considered to represent the North Atlantic. Because the title is already as long and specific as it can be, then at least qualify this aspect prominently in the paper, certainly in the abstract as well as the conclusions.

Changes have been made to the title in order to better reflect the regional nature of the proxy record: “*Multicentennial Record of Labrador Sea Primary Productivity and Sea-Ice Variability Archived in Coralline Algal Ba/Ca*”.

All references to productivity in the North Atlantic has been changed to the Labrador Sea or Northwest Atlantic.

Please note that we have analyzed crustose coralline algae, which is distinct from corals.

C. Data & methodology: *The data and methods are appropriate, and are reasonably well presented in the main manuscript and in the Suppl. Information.*

One aspect that needs to be clarified is:

(2) There are differences apparent in the various sea ice data records (Newfoundland sea ice extent, Fram Strait sea ice export, and the paleo proxy of Halfar et al.) While all show multidecadal variability, there are differences, e.g., in mid-to-late 20th century, where the Ba/Ca peak doesn't appear to correspond well with the Newfoundland ice or the paleo proxy sea ice indicator (Fig. 3 a, b, c). It does however match well with the Fram Strait sea ice export (Fig. 3 d). The authors need to further address this issue, and offer an explanation - maybe the Newfoundland record and/or coral-based sea-ice proxy does not closely reflect sea ice in parts of the record. It is presumably not an issue of chronology nor resolution, as these records are all annual or better - or is there something going on in the Ba/Ca record in that respect?

The Newfoundland sea-ice extent observations were reconstructed from seasonal ice charts (January to April; 1810 – present; <http://archive.nrc-cnrc.gc.ca/eng/ibp/iot/research/ice-databases.html>) depicting ice conditions off the east coast of Newfoundland and the Grand Banks. It is possible that the peaks in the Newfoundland sea-ice extent record did not correspond exactly with the peaks in the algal Ba/Ca due to the very large distance (~1,300 km) between the algal collection site and Newfoundland/Grand Banks region. The Newfoundland sea-ice extent record, and any correlations to that record have thus been removed from the revised manuscript.

The algal Ba/Ca record reported in this paper was obtained from a single location in central Labrador, Canada, while the marine paleo sea-ice proxy reconstruction of Halfar et al. 2013 was averaged from different sites between Labrador and northern Baffin Islands, Canada. Hence, regional differences can be expected. Additional clarification and explanation for the differences

between the algal Ba/Ca and the paleo sea-ice proxy record has been included in the discussions section (line 269 – 275).

D. Statistics and treatment of uncertainties: *There is well-reasoned and correct application of statistics, including correlation analysis that is adjusted for considerable temporal autocorrelation in each of the data records compared. The SSA and the frequency domain spectral analysis appear correct including confidence Levels.*

E. Conclusions: *The robustness, validity and reliability depends to some degree on addressing points (1) and (2) above.*

F. Suggested improvements: *Specific issues in addition to the points above:*

(3) Pg. 2: "This ongoing loss in Arctic sea-ice cover has also led to an increased export of drift ice and freshwater out of the Arctic Ocean through the Fram Strait." This is not necessarily true as stated. There may be an association between the two, but the relationship can even be the opposite to this statement: that is, an increase in export of sea ice from the Arctic Ocean through the Fram Strait can itself lead to decreased sea-ice cover in the Arctic. Case in point: the then-record sea ice minimum in 2007 that was largely due to anomalously high ice export.

For the purposes of this paper, we do not feel that it is not necessary to discuss if the melting Arctic sea ice leads to increased export, or if the increased export leads to a decline in Arctic ice cover. Hence, we have reworded our sentence to: "This ongoing loss in Arctic sea-ice cover has also been associated with an increased export of drift ice and freshwater out of the Arctic Ocean through the Fram Strait, and into the North Atlantic via the East Greenland Current." (line 49 – 51)

(4) Pg. 12. "This multicentennial record of coralline algal Ba/Ca ratios indicates that the recently observed productivity increase in the Subarctic North Atlantic is unprecedented in the last 365 years." The indicator of primary productivity does indeed have its highest value in the most recent years, but the _increase_ (rate of change) is not unusual, so please re-phrase. And it could be mentioned that only very recently have the values exceeded those seen in the mid-20th century, associated with the culmination of the Early 20th Century Warming and a positive AMO state.

This is a very good point. Clarification of this aspect has been added to the discussion section of the revised manuscript, stating that following: "While similar rates of decline have been observed in earlier parts of the algal Ba/Ca record (e.g. 1650 – 1700, 1910 – 1940), it is only recently that productivity has reached values exceeding those observed in the mid-20th century. These values signify the highest levels of productivity since the mid-17th century, which may have resulted from the culmination of a strongly positive AMO state superimposed on the long-term 20th century warming trend." (line 280 – 285)

G. References: *The Reference list is appropriate and excellent. No additions or deletions recommended.*

H. Clarity and context: *The presentation quality is consistently high, including the abstract, introduction and conclusions. The figures are also appropriate in number, information content and are of high quality.*

Reviewer #3 (Remarks to the Author):

The manuscript "Multicentennial Record of North Atlantic Primary Productivity and Sea-Ice Variability Archived in Coralline Algal Ba/Ca" by Chan et al. is a very well written description of a paleo-proxy study of changes in primary production in the Arctic over the last 500 years. I am not a paleo-oceanographer so I won't comment on the methods used, although they appear to be consistent with other papers I have read on similar topics. The results are very interesting and important and well worth publishing in Nature Communications.

I only have a few comments that may need to be addressed by the authors.

It is not clear how solidly it has been established that the Ba/Ca ratio is a proxy for ocean production. I understand the logic employed, but it would be useful for the authors to provide some recent evidence that this proxy has been used reliably in the way the authors use it. The most recent of the three papers they cite is from 1988. I would expect that a good proxy would have been used more often and more recently than that.

We have now included additional and more recent citations of the applications of Ba/Ca as a paleoproxy in the discussion. "Previously, barium-to-calcium trace element ratios have been used for the reconstruction of coastal runoff, stratification, and/or upwelling depending on the source of barium in surface waters, which in turn is largely determined by the setting of the sampling region (Lea et al., 1989; Montaggioni et al., 2006; Sinclair and McCulloch, 2004, Fleitman et al., 2007; LaVigne, 2016)." (Line 240 – 242)

This investigation is the first to demonstrate the Ba/Ca proxy as a productivity proxy related to sea-ice variability. In this revised version of the manuscript, we have verified the proxy against a carbon isotope time series - a commonly accepted productivity proxy (line 119 – 133; also please see above).

Chlorophyll is not the same as primary production and should not be treated as such. This is especially important for their study since increases in primary production in recent years have been attributed to lower sea ice cover and longer growing seasons, not to higher chlorophyll

concentrations. Phytoplankton are growing for a longer period of time, but not necessarily attaining higher biomass.

This aspect was addressed in the revised version of the manuscript, by the following: “Increasing productivity has been linked to larger open water areas (providing suitable ice-free habitats for phytoplankton growth) and longer open water seasons (the timing between spring melt and fall freeze-up - which determines the length of the plankton growing season)⁹.” (line 255 – 257)

I was also left wondering if the growth of coralline algae itself can affect the Ba/Ca and Mg/Ca ratios. This was not addressed directly, although the authors allude to the fact that they record seawater ratios accurately. Also, the authors note a negative correlation between Ba/Ca (a proxy for production) and Mg/Ca (a proxy for ice cover) ratios. Considering these two proxies are clearly not independent (both come from the coralline algae and both contain Ca), is this a valid exercise? It would also be useful for the authors to provide the basis for using Mg/Ca as a proxy for ice cover.

It was not stated in the manuscript that Mg/Ca ratios are used as a proxy for ice cover. Instead, as described in an earlier paper (Halfar et al. 2013), the sea-ice proxy is a combination of Mg/Ca ratios and algal growth increment widths (calibrated against the satellite record). We have clarified this aspect in the results section (line 193 – 197).

Similarly, the paper did not state a negative relationship between Ba/Ca and Mg/Ca, but rather a negative relationship between Ba/Ca and the sea-ice proxy (combined Mg/Ca and growth). This is explained in the following section: “The coralline algal Ba/Ca record correlates negatively to the combined proxy sea-ice record (Fig. 3b; 1646 – 2009; 10 year mean; $n = 317$, $r = -0.75$, $p_{adj} < 0.001$), such that increases in sea-ice proxy values (related to warming, less sea-ice, and increased light levels) are associated with decreases in algal Ba/Ca (resulting from increased primary productivity).” (line 197 – 200)

Growth shows no significant relationship to Mg/Ca or Ba/Ca.

Reviewers' comments:

Reviewer #1 (Remarks to the Author):

Dear editor,

this paper is certainly improved over the original submission via the addition of the d13C to Ba/Ca validation. This new data does, however, indicate that Ba/Ca is not only recording productivity and thus the approach does not support the conclusions drawn.

Overview and Summary of the key results

This paper investigates historic marine productivity using reconstructions from coralline algal-derived Ba/Ca. The paper makes some interesting links between past sea ice and productivity which is a major research question and I commend the authors on this. However, the new Ba/Ca to d13C validation demonstrates that Ba/Ca is not only reconstructing productivity at annual time scales, which are the time scales relevant to seasonal ice melt (see below). Ba/Ca can also record fresh water runoff (e.g Bahr et al 2013 EPSL, Hetzinger et al 2013 Sci Rep) so for the Ba/Ca productivity proxy to work, there must be no freshwater runoff in the area.

Data & methodology: validity of approach, quality of data, quality of presentation

There is a good match in trend between d13C and Ba/Ca at decadal time scales (Fig 2 and Fig S1), but at shorter time scales the two time series do not co-vary (e.g 1925 to 1945). While decadal correlation statistics are given (line 126), no annual correlation statistics between d13C and Ba/Ca are given. This annual resolution mismatch (Fig S1) has implications for what is driving the Ba/Ca at sub-decadal time scales; annual sea ice melt drives annual productivity (via light and nutrients) and this is the resolution at which Ba/Ca and d13C should thus be considered.

While there is decadal correlation between Ba/Ca with C isotopes; this likely occurs as the same large scale climate variables are controlling terrestrial discharge are also controlling longer term trends in sea ice melt (productivity).

Classifying the collection site as "offshore" thus avoiding terrestrial runoff is not appropriate. Kingitok Island (which one, there are several next to each other?) is part of a chain of hundreds of islands just off the Labrador coast (perpendicular distance to the coast is more like 5-10km). The island chain nature and the proximity to the coast classify the areas as strictly coastal and not offshore (that would need to be 100s of km from the coast past the shelf) thus they are under direct influence of any terrestrial runoff. Indeed there is freshwater discharge from Labrador to the coastal area including the Fraser River near Nain along with dispersed discharge during spring.

For the sampling location the drivers of Ba/Ca at annual time scales are thus very likely 1) the influence of terrestrial runoff (which overprints any productivity signal), 2) productivity and 3) natural variability in the proxy system.

Appropriate use of statistics and treatment of uncertainties

Corrections between d13C and Ba/Ca at annual time scales are needed to determine if Ba/Ca is recording productivity or productivity and terrestrial runoff (which could not be decoupled using this proxy).

Clarity and context: lucidity of abstract/summary, appropriateness of abstract, introduction and conclusions

Line 44: remove "etc".

Line 164 -166: Chlorophyll a in ocean colour images near the coastal and shelf zone integrates three signals 1) chlorophyll a, 2) riverine-derived CDOM and 3) riverine-derived sediment. This is why most studies do not include near coastal/shelf data (See Arrigo et al 2011). This should be considered in the context of what the Ba may be recording.

Reviewer #2 (Remarks to the Author):

Dear Editor and Authors,

Regarding the manuscript, " Multicentennial Record of Labrador Sea Primary Productivity and Sea-Ice Variability Archived in Coralline Algal Ba/Ca".":

The authors have responded to each of the 3 or 4 concerns that this reviewer expressed in the initial review. These were in regards to the regional specificity (Labrador Sea rather than North Atlantic) of the variability and trends in primary productivity and sea ice, and a couple of concerns about the stated correlations amongst the historical sea ice records.

I am satisfied with the authors' responses and accordingly the changes made in the revised manuscript. Moreover, the quality of the manuscript it is substantially improved from the initial version. These major improvements in justifying the scientific results and interpretation - and clarity of presentation - derive primarily from the responses to the other reviewers #1 and #3, and/or initiated by the authors themselves.

I recommend acceptance for publication in Nature Communications.

There are a couple of minor edits to address:

Pg 1. Line 23. Abstract. "Accelerated warming and melting of Arctic sea-ice has been associated with significant increases in satellite-based phytoplankton productivity in recent years." As worded, the phytoplankton productivity is satellite-based, rather than the phytoplankton productivity (and sea ice) being estimated from satellite observations. Please reword, e.g., "Accelerated warming and melting of Arctic sea-ice has been associated with significant increases in phytoplankton productivity in recent years, as estimated from satellites", or similar.

Pg. 19, line 45. The Comiso (2012) reference to recent sea ice trends is fine, although alternatively one could refer to the latest (2016) estimates from the near-real-time sea ice data and analysis on the National Snow and Ice Data Center website. I believe that the trends of the recentmost years are essentially the same as stated, so using Comiso (2012) is fine. However, please correct the title text - the words in the title should not be capitalised: Comiso, J. C. Large decadal decline of the arctic multiyear ice cover. *J. Climate* 25, 1176-1193 (2012).

Reviewer #3 (Remarks to the Author):

The authors have successfully addressed all of my previous comments.

Reviewers' comments:

Reviewer #1 (Remarks to the Author):

Dear editor,

this paper is certainly improved over the original submission via the addition of the d13C to Ba/Ca validation. This new data does, however, indicate that Ba/Ca is not only recording productivity and thus the approach does not support the conclusions drawn.

Overview and Summary of the key results

This paper investigates historic marine productivity using reconstructions from coralline algal-derived Ba/Ca. The paper makes some interesting links between past sea ice and productivity which is a major research question and I commend the authors on this. However, the new Ba/Ca to d13C validation demonstrates that Ba/Ca is not only reconstructing productivity at annual time scales, which are the time scales relevant to seasonal ice melt (see below). Ba/Ca can also record fresh water runoff (e.g Bahr et al 2013 EPSL, Hetzinger et al 2013 Sci Rep) so for the Ba/Ca productivity proxy to work, there must be no freshwater runoff in the area.

The reviewer is certainly correct in arguing that freshwater runoff can be a significant source of barium. As the reviewer indicates, the link between Ba/Ca ratios and freshwater runoff has been made in a number of studies (including our own previous work off Newfoundland – which shows a negative correlation between Ba and freshwater - Hetzinger et al., 2013). However, in the case of our present north-central Labrador location, freshwater-derived sources do not contribute significant quantities of barium for a number of reasons:

A) Along the Labrador coast, freshwater runoff is limited (mean average runoff 600 - 700mm, from Newfoundland and Labrador Department of Environment and Conservation - The Hydrology of Labrador:

http://www.env.gov.nl.ca/env/waterres/reports/hydrology_lab/hydrol_lab_chap2.pdf.

B) Oceanographic monitoring closest to our sample collection site has shown the highest salinity recorded in comparison to the rest of the North American coast (offshore salinity station 5 = 33.7 PSU, Fig 3 in Yamamoto-Kawai, 2010 Journal of Marine Research) – which indicates that dilution by freshwater runoff is likely to be minimal in this region. This is supported by other studies which indicate the minimal contribution of river runoff to the Labrador shelf (Tan and Strain, 1996, Journal of Geophysical Research; Myers et al., 1989, Atmosphere-Ocean).

C) Measurements of seawater barium along the Labrador coastline has indicated low concentrations of Ba in surface waters (Fig 3, 4b, Yamamoto-Kawai, 2010 Journal of Marine Research). This indicates that even if there was runoff reaching our sampling site, it would contain minor quantities of barium.

D) In addition, the sampling site is constantly bathed by the high-volume southward flow of the inshore branch of the Labrador Current (estimated at 0.8 Sverdrup and predominantly Baffin Bay origin) which would overwhelmingly dilute any runoff added to the system via the small Labrador river systems. In addition, winds from the northwest induce onshore Ekman transport as evidenced by a strong onshore flow (10-20 cm/s) throughout the water column (Han et al., 2008, Journal of Geophysical Research).

In summary, the regional setting of Labrador is very different from for example the Bahr et al., 2013 paper mentioned by reviewer #1, that studies Orinoco-river derived Ba/Ca in planktonic foraminifera. The Orinoco is the third largest river in the world with massive freshwater discharge during the wet season. The Orinoco flows through steep weathering terrain where rocks from the Andes and Llanos are eroded, leading to high amounts of suspended and dissolved material containing copious amounts of Ba.

Thus, the interpretation of the Ba/Ca proxy for freshwater runoff is governed by the regional setting. In the absence of steep weathering terrain to deliver Ba-rich suspended and dissolved

material, low annual runoff, and distant location to major rivers, the influence of freshwater runoff (and hence Ba contamination) is expected to be insignificant.

In order to clarify this important aspect for the reader, we have added the following section to our revised manuscript (Lines 136-146 and lines 251-253):

“Though it is possible for terrestrial runoff to overprint the barium productivity signal, oceanographic measurements of surface waters taken directly off the coast of Labrador show low barium and high salinity levels resulting from only small quantities of local freshwater input with low barium concentrations (station 5 from Yamamoto-Kawai et al.,³⁶. In addition, the sampling site is constantly bathed by the inshore branch of the southward-flowing Labrador Current, which, with a flow of 0.8 Sverdrup¹¹, dilutes any runoff from the numerous but small river systems along the Labrador coastline. In fact, the Labrador coastline experiences an average runoff of only 600-700mm per year³⁷. Therefore, the dominant source of Ba depletion is likely to be associated with the drawdown of barium through intense biological scavenging as a result of primary production, rather than from terrestrial freshwater inputs off coastal Labrador.”

“As mentioned in the above, minor contributions of local runoff (characterized by low barium concentrations) are largely overprinted by the high-volume flow of the inshore branch of the Labrador Current.”

Data & methodology: validity of approach, quality of data, quality of presentation

There is a good match in trend between d13C and Ba/Ca at decadal time scales (Fig 2 and Fig S1), but at shorter time scales the two time series do not co-vary (e.g 1925 to 1945). While decadal correlation statistics are given (line 126), no annual correlation statistics between d13C and Ba/Ca are given. This annual resolution mismatch (Fig S1) has implications for what is driving the Ba/Ca at sub-decadal time scales; annual sea ice melt drives annual productivity (via

light and nutrients) and this is the resolution at which Ba/Ca and $\delta^{13}\text{C}$ should thus be considered.

We agree that the statistics for the Ba/Ca and $\delta^{13}\text{C}$ relationship should be demonstrated on annual timescales in order to match the resolution of annual ice melt and productivity. We now show that in fact, there are significant correlations between $\delta^{13}\text{C}$ and Ba/Ca on annual timescales (1870-1960; $r = -0.53$, $p < 0.001$). Annual correlations have now been added to the manuscript (line 126-133 and Supplementary Fig.1):

“Both the annually-resolved and decadal-smoothed algal Ba/Ca time series show a close correspondence to the algal $\delta^{13}\text{C}$ record (1870-1960; annual mean; $r = -0.53$; $p < 0.001$; Supplementary Fig. 1) (Fig. 2 a, b; 1870-1960; 10 year mean; $r = -0.88$, $p_{\text{adj}} = 0.0038$). Significance levels were adjusted for loss in degrees of freedom using a lag-1 test that takes into account the temporal autocorrelation of the underlying time series³⁶ (p_{adj} = adjusted for loss of degrees of freedom). The high degree of correlation between both the Ba/Ca and $\delta^{13}\text{C}$ time series prior to 1960 indicates that algal Ba/Ca ratios at this site can be used as a proxy for productivity that does not suffer from the anthropogenically-induced carbon isotope decline.”

The minor mismatch between $\delta^{13}\text{C}$ and Ba/Ca in 1925 and 1945 may be accounted for by proxy variability between the two different samples from which the records were obtained. This aspect has been further emphasized in the results section (line 133-136):

“Minor departures observed between the annually-resolved algal Ba/Ca and $\delta^{13}\text{C}$ timeseries may be explained by the natural variability within the proxy system, resulting from the records being obtained from two different specimens collected from nearby sites.”

While there is decadal correlation between Ba/Ca with C isotopes; this likely occurs as the same large scale climate variables are controlling terrestrial discharge are also controlling longer term trends in sea ice melt (productivity).

As discussed in detail above, oceanographic measurements indicate that terrestrial discharge plays only a minor role at our sampling site and contains low concentrations of barium. The long-term climate trends influencing sea ice melt and productivity (hence algal Ba/Ca via biological scavenging) are described in the AMO section.

Classifying the collection site as “offshore” thus avoiding terrestrial runoff is not appropriate. Kingitok Island (which one, there are several next to each other?) is part of a chain of hundreds of islands just off the Labrador coast (perpendicular distance to the coast is more like 5-10km). The island chain nature and the proximity to the coast classify the areas as strictly coastal and not offshore (that would need to be 100s of km from the coast past the shelf) thus they are under direct influence of any terrestrial runoff. Indeed there is freshwater discharge from Labrador to the coastal area including the Fraser River near Nain along with dispersed discharge during spring.

Reviewer #1 is correct. Due to the short distance of the sampling site to shore, the site should be classified as “coastal”. Offshore classification of the sample collection site has now been removed (line 111-115).

*“Two living specimens of the alga *Clathromorphum compactum* were collected: Sample Ki1 in July 2011 at 15-17 m water depth, and sample 2013-15-4 in July 2013 at 17 water depth via SCUBA off the east coast of Eastern Kingitok Island in Labrador, Canada approximately 15 km offshore from central Labrador, Canada (55°26'6.58"N, 59°51'55.57"W) (Fig. 1 a, b) (for detailed description of sampling site, see Adey et al.,³⁵).*”

However, as discussed above, the sampling location is situated within the pathway of flow of the inshore branch of the Labrador Current (Figure 2 from Lazier and Wright, 1993, American Meteorological Society, and Figure 7, Han et al., 2005, Continental Shelf Research), which transports a large amount of North Atlantic oceanic water southwards directly along the Labrador coast, thus essentially removing the influence of any significant contribution of terrestrial runoff.

The Fraser River near Nain (a small river extending 105 km inshore, not to be confused with the very large Fraser River in British Columbia) is about 155 km north of our sample collection site. Therefore it is highly unlikely that the Fraser River will have a significant effect on the water composition at Kingitok.

For the sampling location the drivers of Ba/Ca at annual time scales are thus very likely 1) the influence of terrestrial runoff (which overprints any productivity signal), 2) productivity and 3) natural variability in the proxy system.

As can be seen from the above 1) can be ruled out, and 2) will contribute the overwhelming signal to our time series. We do however, agree that there is natural variability in the proxy system 3) – as demonstrated by the short term mismatch between the annual Ba and $\delta^{13}C$ taken from two different specimens. The latter point is now emphasized in the revised version of the manuscript (line 133-136, please see above).

Appropriate use of statistics and treatment of uncertainties

Correlations between $\delta^{13}C$ and Ba/Ca at annual time scales are needed to determine if Ba/Ca is recording productivity or productivity and terrestrial runoff (which could not be decoupled using this proxy).

As shown in the above, there are significant correlations between $\delta^{13}\text{C}$ and Ba/Ca on annual timescales. Annual correlations have been added to the manuscript (line 126-133, please see above).

Clarity and context: lucidity of abstract/summary, appropriateness of abstract, introduction and conclusions

Line 44: remove “etc”.

“.etc” has been removed from the end of the sentence on (line 41-44).

“This blooming process is dependent upon physical properties of the surface seawater (ex. temperature, stratification, mixed-layer depth, light levels, and nutrient availability) that are directly modified by climatological factors (ex. solar radiation, cloud cover, wind mixing, and upwelling)⁴.”

Line 164 -166: Chlorophyll a in ocean colour images near the coastal and shelf zone integrates three signals **1**) chlorophyll a, **2**) riverine-derived CDOM and **3**) riverine-derived sediment. This is why most studies do not include near coastal/shelf data (See Arrigo et al 2011). This should be considered in the context of what the Ba may be recording.

The spatial region of SeaWiFS data for comparison to algal Ba/Ca has now been adjusted to ensure that no coastal and/or land areas are included (to remove the influence of coastlines and riverine-derived sediments on SeaWiFS data). (please see updated Fig. 1).

Reviewer #2 (Remarks to the Author):

Dear Editor and Authors,

Regarding the manuscript, " Multicentennial Record of Labrador Sea Primary Productivity and Sea-Ice Variability Archived in Coralline Algal Ba/Ca".":

The authors have responded to each of the 3 or 4 concerns that this reviewer expressed in the initial review. These were in regards to the regional specificity (Labrador Sea rather than North Atlantic) of the variability and trends in primary productivity and sea ice, and a couple of concerns about the stated correlations amongst the historical sea ice records.

I am satisfied with the authors' responses and accordingly the changes made in the revised manuscript. Moreover, the quality of the manuscript it is substantially improved from the initial version. These major improvements in justifying the scientific results and interpretation - and clarity of presentation - derive primarily from the responses to the other reviewers #1 and #3, and/or initiated by the authors themselves.

I recommend acceptance for publication in Nature Communications.

There are a couple of minor edits to address:

Pg 1. Line 23. Abstract. "Accelerated warming and melting of Arctic sea-ice has been associated with significant increases in satellite-based phytoplankton productivity in recent years." As worded, the phytoplankton productivity is satellite-based, rather than the phytoplankton productivity (and sea ice) being estimated from satellite observations. Please reword, e.g., "Accelerated warming and melting of Arctic sea-ice has been associated with significant increases in phytoplankton productivity in recent years, as estimated from satellites", or similar.

The word "estimates" has been added to the sentence (line 23-24):

“Accelerated warming and melting of Arctic sea-ice has been associated with significant increases in satellite-based phytoplankton productivity estimates in recent years.”

Pg. 19, line 45. The Comiso (2012) reference to recent sea ice trends is fine, although alternatively one could refer to the latest (2016) estimates from the near-real-time sea ice data and analysis on the National Snow and Ice Data Center website. I believe that the trends of the recent most years are essentially the same as stated, so using Comiso (2012) is fine. However, please correct the title text - the words in the title should not be capitalised: Comiso, J. C. Large decadal decline of the arctic multiyear ice cover. *J. Climate* 25, 1176-1193 (2012).

Capitalization has been removed from the words in the title (line 455-456).

*Comiso, J. C. Large decadal decline of the Arctic multiyear ice cover. *J. Climate* 25, 1176-1193 (2012).*

Reviewer #3 (Remarks to the Author):

The authors have successfully addressed all of my previous comments.

Reviewers' comments:

Reviewer #1 (Remarks to the Author):

1. Please add a plot showing annual Ba/Ca against $\delta^{13}\text{C}$ (a cross plot of Fig S1) to show the variability in the relationship in the context of the overall variability reconstructed. This will allow the reader to assess the signal to noise ratio in the context of the reconstructed change.
2. Add time series data on annual salinity variation (e.g. from Yamamoto Kawai) to enable to reader to get a feel for the freshwater fluctuations. This could be added to Fig 2.

Reviewer's Comments:

Reviewer #1 (Remarks to the Author):

1. Please add a plot showing annual Ba/Ca against $\delta^{13}\text{C}$ (a cross plot of Fig S1) to show the variability in the relationship in the context of the overall variability reconstructed. This will allow the reader to assess the signal to noise ratio in the context of the reconstructed change.

We have followed Reviewer #1's suggestion, and have added a cross plot of Supplementary Fig. 1 - Annual Ba/Ca against $\delta^{13}\text{C}$ (Line 128; Supplementary Fig. 1b).

2. Add time series data on annual salinity variation (e.g. from Yamamoto Kawai) to enable to reader to get a feel for the freshwater fluctuations. This could be added to Fig 2.

As requested, we have added a comparison of algal Ba/Ca to an annual salinity time series (going back to year 1900). Unfortunately the Yamamoto-Kawai dataset did not contain enough data points for meaningful comparison. Therefore, we have obtained annually-resolved gridded sea surface salinity for the region depicted in Fig. 1. Correlations are in agreement with the literature, and further support our findings, demonstrating no statistically significant relationship between algal Ba/Ca and salinity. We have decided to present this annually resolved figure in the supplementary section rather than to include it into Fig. 2 because all of the other timeseries were shown as 10 year means (in order to minimize natural variability within the proxy system) (Lines 140-144; Supplementary Fig. 2).

Reviewers' comments:

Reviewer #1 (Remarks to the Author):

The authors have now provided the two plots requested. $\delta^{13}\text{C}$ is a productivity proxy while Ba/Ca records salinity and also productivity. If Ba/Ca was only being influenced by productivity at this site, then Ba/Ca plotted against $\delta^{13}\text{C}$ should give a nice, tight, relationship. That relationship is plotted in S1B; you will see a variable relationship between the two (if the dotted line was not there it would be tricky to work out if there was a relationship). For example, for a defined value of $\delta^{13}\text{C}$ (e.g. 1.5) the Ba/Ca nearly varies across the whole range of Ba/Ca values recorded (2.1×10^{-5} to 3.0×10^{-5}) (ie. Very high noise). This means either 1) Ba/Ca is not recording productivity (annual Ba/Ca does poorly correlates with annual $\delta^{13}\text{C}$) and / or 2) Ba/Ca is recording productivity as well as something else giving the very large range of values (likely salinity, the trend in the interpolated data plot of salinity to Ba/Ca is very similar (sup fig 2)).

In the context of reconstructing productivity, this means for any annual value of Ba/Ca, the error on that value is equal to range of productivity measured over the whole time series (i.e. the noise is larger than the change being reconstructed). Thus, it is not possible to tell if there is a relationship between Ba/Ca and productivity.

Given this, it is not possible to tell what is driving the Ba/Ca patterns so they are not reconstructing only productivity. At present, as their data does not support their conclusions I cannot suggest publication of the paper.

Below is our point-by-point response to the comments made by the reviewer (*in italics*):

Reviewer #1 (Remarks to the Author NCOMMS-16-02601C):

The authors have now provided the two plots requested. $d13C$ is a productivity proxy while Ba/Ca records salinity and also productivity. If Ba/Ca was only being influenced by productivity at this site, then Ba/Ca plotted against $d13C$ should give a nice, tight, relationship. That relationship is plotted in S1B; you will see a variable relationship between the two (if the dotted line was not there it would be tricky to work out if there was a relationship). For example, for a defined value of $d13c$ (e.g. 1.5) the Ba/Ca nearly varies across the whole range of Ba/Ca values recorded ($2.1e-5$ to $3.0e-5$) (ie. Very high noise). This means either 1) Ba/Ca is not recording productivity (annual Ba/Ca does poorly correlates with annual $d13C$) and / or 2) Ba/Ca is recording productivity as well as something else giving the very large range of values (likely salinity, the trend in the interpolated data plot of salinity to Ba/Ca is very similar (sup fig 2). In the context of reconstructing productivity, this means for any annual value of Ba/Ca, the error on that value is equal to range of productivity measured over the whole time series (i.e. the noise is larger than the change being reconstructed). Thus, it is not possible to tell if there is a relationship between Ba/Ca and productivity.

Given this, it is not possible to tell what is driving the Ba/Ca patterns so they are not reconstructing only productivity. At present, as their data does not support their conclusions I cannot suggest publication of the paper.

Response to Reviewer's Comments:

Our barium record shows a statistically significant relationship to carbon isotopes while no statistical relationship exists with salinity (as has been shown in the supplementary information section of the most recently submitted manuscript version), and as demonstrated below (displayed as **Supplementary Figure 1 a, b and c** in the revised manuscript):

Supplementary Figure 1 | Relationship between Ba/Ca and $\delta^{13}\text{C}$. a) Annual and 10 year mean algal Ba/Ca ratios were derived from sample Kil and $\delta^{13}\text{C}$ values from sample 2013-15-4 (plotted inversely on the secondary x-axis). The post-1960s departure to negative $\delta^{13}\text{C}$ values is due to the anthropogenic introduction of isotopically light fossil fuels into the biosphere ("Suess Effect"). A $\delta^{13}\text{C}$ decline starting ~1970 has commonly

been observed in marine proxy archives throughout the North Atlantic. **b)** Annually resolved; and **c)** Decadally-smoothed Ba/Ca plotted against $\delta^{13}\text{C}$ for the period prior to the Suess Effect displayed as scatter plots. Significance levels were adjusted for loss in degrees of freedom using a lag-1 test that takes into account the temporal autocorrelation of the underlying time series (p_{adj} = adjusted for loss of degrees of freedom).

We agree that there is some scatter in the annual and 10 year mean plots of algal Ba/Ca vs. $\delta^{13}\text{C}$. However, a significant coupling exists between the two proxies (**Supplementary Figure 1 a, b and c**), despite the fact that Ba/Ca and $\delta^{13}\text{C}$ were obtained from two separate algal specimens collected from two different nearby sites. As shown in the supplementary section of the last submission, the annual correlations between Ba/Ca and $\delta^{13}\text{C}$ covary over a period of 100 years prior to the 1970s (**Supplementary Figure 1a**). The algal Ba/Ca to $\delta^{13}\text{C}$ relationship becomes even more prominent at decadal timescales (shown by 10 year means; $r_{10 \text{ year mean}} = -0.89$, $p_{\text{adj}} = 0.003$; **Supplementary Figure 1 a and c**). Decadal means have thus been used in the main figure of the manuscript (e.g. **Fig. 2**) to highlight our comparisons between algal Ba/Ca and various observational and proxy records of productivity and sea-ice. Annual correlations over the last 140 years show the unprocessed data set (**Supplementary Figure 1a**), and are displayed in the supplementary information section to provide additional information to the reader. Through the examination of decadal-scale trends, the main goal of our paper is to reveal broad trends and mechanistic links between climatic changes, sea-ice cover, and primary productivity - rather than to provide exact quantitative measures of productivity.

We have now included the correlation between Ba/Ca and $\delta^{13}\text{C}$ as line and scatterplots in the supplementary information section - **Supplementary Figure 1a, b and c** and captions, and (lines 127 – 129) of the revised manuscript showing a statistically significant relationship between the barium record to carbon isotopes.

Finally, we disagree with the reviewer's final comments in saying that:

"For example, for a defined value of $d13c$ (e.g. 1.5) the Ba/Ca nearly varies across the whole range of Ba/Ca values recorded ($2.1e-5$ to $3.0e-5$) (ie. Very high noise). This means either 1) Ba/Ca is not recording productivity (annual Ba/Ca does poorly correlates with annual $d13C$) and / or 2) Ba/Ca is recording productivity as well as something else giving the very large range of values (likely salinity, the trend in the interpolated data plot of salinity to Ba/Ca is very similar."

In response to **1) Ba/Ca correlates poorly with $\delta^{13}\text{C}$** - The correlation is statistically significant and strong, particularly on decadal timescales (e.g. $r_{10 \text{ year mean}} = -0.89$, $p_{\text{adj}} = 0.003$). Furthermore, the spread of Ba/Ca values for a fixed value of $\delta^{13}\text{C}$ (e.g. 1.5) does not range from ($2.1e-5$ to $3.0e-5$) as implied by the reviewer. The spread of Ba/Ca values for a fixed value of 1.5 $\delta^{13}\text{C}$ in fact ranges from ($2.2e-5$ to $2.6e-5$) with the standard error of Ba/Ca on the order of $\pm 0.2e-5$. This clearly shows that, contrary to the reviewer's assertion, the Ba/Ca for a fixed $\delta^{13}\text{C}$ value do not span the entire range of values. Indeed the reviewer's estimate of the standard error was twice as large as what is implied by the data.

In response to **2) Ba/Ca is likely recording salinity** - Based on the figures shown below (displayed as **Supplementary Figure 2 a, b and c** in the revised manuscript) we have reason to strongly disagree with the above statement that *"the trend in the interpolated data plot of salinity to Ba/Ca is very similar"* [to the trend of Ba/Ca to $\delta^{13}\text{C}$].

a) Annual Ba/Ca vs. Salinity (1900 – 2009; $r = -0.19$, $p_{adj} = 0.29$)

10 year mean Ba/Ca vs. Salinity (1900 – 2009; $r = -0.35$, $p_{adj} = 0.32$)

b) Annual Ba/Ca vs. Salinity

(1900 – 2009; $n = 110$, $r = -0.19$, $p = 0.052$, $p_{adj} = 0.29$)

c) 10 year mean Ba/Ca vs. Salinity

(1900 – 2009; $n = 101$, $r = -0.35$, $p_{adj} = 0.32$)

Supplementary Figure 2 | Annual Ba/Ca versus salinity. a) Annual and 10 year mean Gridded salinity data (Met Office Hadley Centre observations dataset EN 4.2.0²) for the rectangular region depicted in Fig. 1 (56.0577 - 58.0133° N; 57.0905 - 61.0455° W) show no significant correlations to algal Ba/Ca. Algal Ba/Ca ratios versus salinity displayed as scatter plots on b) Annual timescales (1900 – 2009; $r = -0.19$, $p > 0.05$; $p_{adj} = 0.29$); and c) Decadal timescales (1900 – 2009; $n = 101$, $r = -0.35$, $p_{adj} = 0.32$). Significance levels were adjusted for loss in degrees of freedom using a lag-1 test that takes into account the temporal autocorrelation of the underlying time series (p_{adj} = adjusted for loss of degrees of freedom).

Although the trend in salinity may seem “similar”, the spread in the values as shown in **Supplementary Figure 2 b and c** are much larger than those in **Supplementary Figure 1 b and c**. Furthermore, a “similar” trend between Ba/Ca and salinity would have resulted in a much higher correlation coefficient that was statistically significant – which is not the case.

The line and scatterplots of Ba/Ca vs. salinity have now been included as **Supplementary Figure 2a, b and c** and captions in the supplementary figures section, and (lines 140 – 146) of the revised manuscript, showing no significant relationship between barium record and salinity.